# A Temporal Kernel Approach for Deep Learning with Continuous-time Information

**Da Xu**
Walmart Labs
Sunnyvale, CA 94086, USA
DaXu5180@gmail.com

**Chuanwei Ruan** *
Instacart
San Francisco, CA 94107, USA
Ruanchuanwei@gmail.com

**Evren Korpeoglu, Sushant Kuamr, Kannan Achan**
Walmart Labs
Sunnyvale, CA 94086, USA
[EKorpeoglu, SKumar4, KAchan]@walmartlabs.com

## Abstract

Sequential deep learning models such as RNN, causal CNN and attention mechanism do not readily consume continuous-time information. Discretizing the temporal data, as we show, causes inconsistency even for simple continuous-time processes. Current approaches often handle time in a heuristic manner to be consistent with the existing deep learning architectures and implementations. In this paper, we provide a principled way to characterize continuous-time systems using deep learning tools. Notably, the proposed approach applies to all the major deep learning architectures and requires little modifications to the implementation. The critical insight is to represent the continuous-time system by composing neural networks with a temporal kernel, where we gain our intuition from the recent advancements in understanding deep learning with Gaussian process and neural tangent kernel. To represent the temporal kernel, we introduce the random feature approach and convert the kernel learning problem to spectral density estimation under reparameterization. We further prove the convergence and consistency results even when the temporal kernel is non-stationary, and the spectral density is misspecified. The simulations and real-data experiments demonstrate the empirical effectiveness of our temporal kernel approach in a broad range of settings.

## 1 Introduction

Deep learning models have achieved remarkable performances in sequence learning tasks leveraging the powerful building blocks from *recurrent neural networks (RNN)* (Mikolov et al., 2010), *long-short term memory (LSTM)* (Hochreiter & Schmidhuber, 1997), *causal convolution neural network (CausalCNN/WaveNet)* (Oord et al., 2016) and attention mechanism (Bahdanau et al., 2014; Vaswani et al., 2017). Their applicability to the continuous-time data, on the other hand, is less explored due to the complication of incorporating time when the sequence is irregularly sampled (spaced). The widely-adopted workaround is to study the discretized counterpart instead, e.g. the temporal data is aggregated into bins and then treated as equally-spaced, with the hope to approximate the temporal signal using the sequence information. It is perhaps without surprise, as we show in Claim 1, that even for regular temporal sequence the discretization modifies the spectral structure. The gap can only be amplified for irregular data, so discretizing the temporal information will almost always

---

*The work was done when the author was with Walmart Labs.

introduce intractable noise and perturbations, which emphasizes the importance to characterize the continuous-time information directly. Previous efforts to incorporate temporal information for deep learning include concatenating the time or timespan to feature vector (Choi et al., 2016; Lipton et al., 2016; Li et al., 2017b), learning the generative model of time series as missing data problems (Soleimani et al., 2017; Futoma et al., 2017), characterizing the representation of time (Xu et al., 2019; 2020; Du et al., 2016) and using neural point process (Mei & Eisner, 2017; Li et al., 2018). While they provide different tools to expand neural networks to coupe with time, the underlying continuous-time system and process are involved explicitly or implicitly. As a consequence, it remains unknown in what way and to what extend are the continuous-time signals interacting with the original deep learning model. Explicitly characterizing the continuous-time system (via differential equations), on the other hand, is the major pursuit of classical signal processing methods such as smoothing and filtering (Doucet & Johansen, 2009; Särkkä, 2013). The lack of connections is partly due to the compatibility issues between the signal processing methods and the auto-differential gradient computation framework of modern deep learning. Generally speaking, for continuous-time systems, model learning and parameter estimation often rely on the more complicated differential equation solvers (Raissi & Karniadakis, 2018; Raissi et al., 2018a). Although the intersection of neural network and differential equations is gaining popularity in recent years, the combined neural differential methods often require involved modifications to both the modelling part and implementation detail (Chen et al., 2018; Baydin et al., 2017).

Inspired by the recent advancement in understanding neural network with Gaussian process and the *neural tangent kernel* (Yang, 2019; Jacot et al., 2018), we discover a natural connection between the continuous-time system and the neural Gaussian process after composing with a temporal kernel. The significance of the temporal kernel is that it fills in the gap between signal processing and deep learning: we can explicitly characterize the continuous-time systems while maintaining the usual deep learning architectures and optimization procedures. While the kernel composition is also known for integrating signals from various domains (Shawe-Taylor et al., 2004), we face the additional complication of characterizing and learning the unknown temporal kernel in a data-adaptive fashion. Unlike the existing kernel learning methods where at least the parametric form of the kernel is given (Wilson et al., 2016), we have little context on the temporal kernel, and aggressively assuming the parametric form will risk altering the temporal structures implicitly just like discretization. Instead, we leverage the Bochner's theorem and its extension (Bochner, 1948; Yaglom, 1987) to first covert the kernel learning problem to the more reasonable spectral domain where we can direct characterize the spectral properties with random (Fourier) features. Representing the temporal kernel by random features is favorable as we show they preserve the existing Gaussian process and NTK properties of neural networks. This is desired from the deep learning's perspective since our approach will not violate the current understandings of deep learning. Then we apply the reparametrization trick (Kingma & Welling, 2013), which is a standard tool for generative models and Bayesian deep learning, to jointly optimize the spectral density estimator. Furthermore, we provide theoretical guarantees for the random-feature-based kernel learning approach when the temporal kernel is non-stationary, and the spectral density estimator is misspecified. These two scenarios are essential for practical usage but have not been studied in the previous literature. Finally, we conduct simulations and experiments on real-world continuous-time sequence data to show the effectiveness of the temporal kernel approach, which significantly improves the performance of both standard neural architectures and complicated domain-specific models. We summarize our contributions as follow.

- We study a novel connection between the continuous-time system and neural network via the composition with a temporal kernel.

- We propose an efficient kernel learning method based on random feature representation, spectral density estimation and reparameterization, and provide strong theoretical guarantees when the kernel is nonstationary and the spectral density is misspeficied.

- We analyze the empirical performance of our temporal kernel approach for both the standard and domain-specific deep learning models through real-data simulation and experiments.

## 2 NOTATIONS AND BACKGROUND

We use bold-font letters to denote vectors and matrices. We use $\mathbf{x}_t$ and $(\mathbf{x}, t)$ interchangeably to denote a time-sensitive event occurred at time $t$, with $t \in \mathcal{T} \equiv [0, t_{\max}]$. Neural networks are denoted

by $f(\boldsymbol{\theta}, \mathbf{x})$, where $\mathbf{x} \in \mathcal{X} \subset \mathbb{R}^d$ is the input where diameter$(\mathcal{X}) \leq l$, and the network parameters $\boldsymbol{\theta}$ are sampled i.i.d from the standard normal distribution at initialization. Without loss of generality, we study the standard L-layer feedforward neural network with its output at the $h^{th}$ hidden layer given by $\boldsymbol{f}^{(h)} \in \mathbb{R}^{d_h}$. We use $\epsilon$ and $\epsilon(t)$ to denote Gaussian noise and continuous-time Gaussian noise process. By convention, we use $\otimes$ and $\circ$ and to represent the tensor and outer product.

## 2.1 UNDERSTANDING THE STANDARD NEURAL NETWORK

We follow the settings from Jacot et al. (2018); Yang (2019) to briefly illustrate the limiting Gaussian behavior of $f(\boldsymbol{\theta}, \mathbf{x})$ at initialization, and its training trajectory under weak optimization. As $d_1, \ldots, d_L \to \infty$, $\boldsymbol{f}^{(h)}$ tend in law to i.i.d Gaussian processes with covariance $\boldsymbol{\Sigma}^h \in \mathbb{R}^{d_h \times d_h}$: $\boldsymbol{f}^{(h)} \sim N(0, \boldsymbol{\Sigma}^h)$, which we refer to as the *neural network kernel* to distinguish from the other kernel notions. Also, given a training dataset $\{\mathbf{x}_i, y_i\}_{i=1}^n$, let $\boldsymbol{f}(\boldsymbol{\theta}^{(s)}) = (f(\boldsymbol{\theta}^{(s)}, \mathbf{x}_1), \ldots, f(\boldsymbol{\theta}^{(s)}, \mathbf{x}_n))$ be the network outputs at the $s^{th}$ training step and $\mathbf{y} = (y_1, \ldots, y_n)$. Using the squared loss for example, when training with infinitesimal learning rate, the outputs follows: $\mathrm{d}\boldsymbol{f}(\boldsymbol{\theta}^{(s)})/\mathrm{d}s = -\boldsymbol{\Theta}(s) \times (\boldsymbol{f}(\boldsymbol{\theta}^{(s)}) - \mathbf{y})$, where $\boldsymbol{\Theta}(s)$ is the *neural tangent kernel* (NTK). The detailed formulation of $\boldsymbol{\Sigma}^h$ and $\boldsymbol{\Theta}(s)$ are provided in Appendix A.2. We introduce the two concepts here because:

**1.** instead of incorporating time to $f(\boldsymbol{\theta}, \mathbf{x})$, which is then subject to its specific structures, can we alternatively consider an universal approach which expands the $\boldsymbol{\Sigma}^h$ to the temporal domain such as by composing it with a time-aware kernel?

**2.** When jointly optimize the unknown temporal kernel and the model parameters, how can we preserve the results on the training trajectory with NTK?

In our paper, we show that both goals are achieved by representing a temporal kernel via random features.

## 2.2 DIFFERENCE BETWEEN CONTINUOUS-TIME AND ITS DISCRETIZATION

We now discuss the gap between continuous-time process and its equally-spaced discretization. We study the simple univariate continuous-time system $f(t)$:

$$\frac{\mathrm{d}^2 f(t)}{\mathrm{d}t^2} + a_0 \frac{\mathrm{d}f(t)}{\mathrm{d}t} + a_1 f(t) = b_0 \epsilon(t). \tag{1}$$

A discretization with a fixed interval is then given by: $f_{[i]} = f(i \times \text{interval})$ for $i = 1, 2, \ldots$. Notice that $f(t)$ is a second-order auto-regressive process, so both $f(t)$ and $f_{[i]}$ are stationary. Recall that the covariance function for a stationary process is given by $k(t) := \text{cov}(f(t_0), f(t_0 + t))$, and the spectral density function (SDF) is defined as $s(\omega) = \int_{-\infty}^{\infty} \exp(-i\omega t) k(t) dt$.

**Claim 1.** *The spectral density function for $f(t)$ and $f_{[i]}$ are different.*

The proof is relegated to Appendix A.2.2. The key takeaway from the example is that the spectral density function, which characterizes the signal on the frequency domain, is altered implicitly even by regular discretization in this simple case. Hence, we should be cautious about the potential impact of the modelling assumption, which eventually motivates us to explicitly model the spectral distribution.

## 3 METHODOLOGY

We first explain our intuition using the above example. If we take the Fourier transform on (1) and rearrange terms, it becomes: $\tilde{f}(i\omega) = \left(\frac{b_0}{(i\omega)^2 + a_0(i\omega) + a_1}\right) \tilde{\epsilon}(i\omega)$, where $\tilde{f}(i\omega)$ and $\tilde{\epsilon}(i\omega)$ are the Fourier transform of $f(t)$ and $\epsilon(t)$. Note that the spectral density of a Gaussian noise process is constant, i.e. $|\tilde{\epsilon}(i\omega)|^2 = p_0$, so the spectral density of $f(t)$ is given by: $s_{\boldsymbol{\theta}_T}(\omega) = p_0 |b_0/((i\omega)^2 + a_0(i\omega) + a_1)|^2$, where we use $\boldsymbol{\theta}_T = [a_0, a_1, b_0]$ to denote the parameters of the linear dynamic system defined in (1). The subscript $T$ is added to distinguish from the parameters of the neural network. The classical Wiener-Khinchin theorem (Wiener et al., 1930) states that the

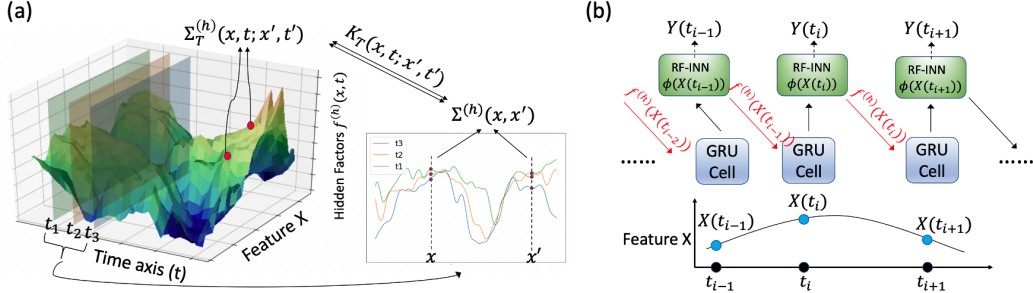

Figure 1: (a). The relations between neural network kernel $\mathbf{\Sigma}^{(h)}$, temporal kernel $K_T$ and the neural-temporal kernel $\mathbf{\Sigma}_T^{(h)}$. (b). Composing a single-layer RNN with temporal kernel where the hidden out from GRU cells become $f^{(h)}(\mathbf{x}, t) \equiv f^{(h)}(\mathbf{x}) \circ \phi(\mathbf{x}, t)$. We use the **RF-INN** blocks to denote the random feature representation parameterized by INN. The optional outputs $y(t)$ can be obtained in a way similar to $f^{(h)}(\mathbf{x}, t)$.

covariance function for $f(t)$, which is a Gaussian process since the linear differential equation is a linear operation on $\epsilon(t)$, is given by the inverse Fourier transform of the spectral density:

$$K_T(t, t') := k_{\boldsymbol{\theta}_T}(t' - t) = \frac{1}{2\pi} \int s_{\boldsymbol{\theta}_T}(\omega) \exp(i\omega t) d\omega, \qquad (2)$$

We defer the discussions on the inverse direction, that given a kernel $k_{\boldsymbol{\theta}_T}(t' - t)$ we can also construct a continuous-time system, to Appendix A.3.1. Consequently, there is a correspondence between the parameterization of a stochastic ODE and the kernel of a Gaussian process. The mapping is not necessarily one-to-one; however, it may lead to a more convenient way to parameterize a continuous-time process alternatively using deep learning models, especially knowing the connections between neural network and Gaussian process (which we highlighted in Section 2.1).

To connect the neural network kernel $\mathbf{\Sigma}^{(h)}$ (e.g. for the $h^{th}$ layer of the FFN) to a continuous-time system, the critical step is to understand the interplay between the neural network kernel and the *temporal kernel* (e.g. the kernel in (2)):

- the neural network kernel characterizes the covariance structures among the hidden representation of data (transformed by the neural network) at any fixed time point;

- the temporal kernel, which corresponds to some continuous-time system, tells how each static neural network kernel propagates forward in time. See Figure 1a for a visual illustration.

Continuing with Example 1, it is straightforward construct the integrated continuous-time system as:

$$a_2(\mathbf{x}) \frac{\mathrm{d}^2 f(\mathbf{x}, t)}{\mathrm{d}t^2} + a_1(\mathbf{x}) \frac{\mathrm{d}f(\mathbf{x}, t)}{\mathrm{d}t} + a_0(\mathbf{x}) f(\mathbf{x}, t) = b_0(\mathbf{x})\epsilon(\mathbf{x}, t), \ \epsilon(\mathbf{x}, t = t_0) \sim N(0, \mathbf{\Sigma}^{(h)}), \ \forall t_0 \in \mathcal{T}, \qquad (3)$$

where we use the neural network kernel $\mathbf{\Sigma}^{(h)}$ to define the Gaussian process $\epsilon(\mathbf{x}, t)$ on the feature dimension, so the ODE parameters are now functions of the data as well. To see that (3) generalizes the $h^{th}$ layer of a FFN to the temporal domain, we first consider $a_2(\mathbf{x}) = a_1(\mathbf{x}) = 0$ and $a_0(x) = b_0(x)$. Then the continuous-time process $f(\mathbf{x}, t)$ exactly follows $f^{(h)}$ at any fixed time point $t$, and its trajectory on the time axis is simply a Gaussian process. When $a_1(\mathbf{x}), a_2(\mathbf{x}) \neq 0$, $f(\mathbf{x}, t)$ still matches $f^{(h)}$ at the initial point, but its propagation on the time axis becomes nontrivial and is now characterized by the constructed continuous-time system. We can easily the setting to incorporate higher-order terms:

$$a_n(\mathbf{x}) \frac{\mathrm{d}^n f(\mathbf{x}, t)}{\mathrm{d}t^n} + \cdots + a_0(\mathbf{x}) f(\mathbf{x}, t) = b_m(\mathbf{x}) \frac{\mathrm{d}^m \epsilon(\mathbf{x}, t)}{\mathrm{d}t^m} + \cdots + b_0(\mathbf{x})\epsilon(\mathbf{x}, t). \qquad (4)$$

Keeping the heuristics in mind, an immediate question is what is the structure of the corresponding kernel function after we combine the continuous-time system with the neural network kernel?

**Claim 2.** *The kernel function for $f(\mathbf{x}, t)$ in (4) is given by: $\Sigma_T^{(h)}(\mathbf{x}, t; \mathbf{x}', t') = k_{\boldsymbol{\theta}_T}(\mathbf{x}, t; \mathbf{x}', t') \cdot \boldsymbol{\Sigma}^{(h)}(\mathbf{x}, \mathbf{x}')$, where $\boldsymbol{\theta}_T$ is the underlying parameterization of $\{a_i(\cdot)\}_{i=1}^n$ and $\{b_i(\cdot)\}_{i=1}^m$ as functions of $\mathbf{x}$. When $\{a_i\}_{i=1}^n$ and $\{b_i\}_{i=1}^m$ are scalars, $\Sigma_T^{(h)}(\mathbf{x}, t; \mathbf{x}', t') = k_{\boldsymbol{\theta}_T}(t, t') \cdot \boldsymbol{\Sigma}^{(h)}(\mathbf{x}, \mathbf{x}')$.*

We defer the proof and the discussion on the inverse direction from temporal kernel to continuous-time system to Appendix A.3.1. Claim 2 shows that it is possible to expand any layer of a standard neural network to the temporal domain, as part of a continuous-time system using kernel composition. The composition is flexible and can happen at any hidden layer. In particular, given the temporal kernel $\mathbf{K}_T$ and neural network kernel $\boldsymbol{\Sigma}^{(h)}$, we obtain the *neural-temporal kernel* on $\mathcal{X} \times \mathcal{T}$: $\boldsymbol{\Sigma}_T^{(h)} = \text{diag}\big(\boldsymbol{\Sigma}^{(h)} \otimes \mathbf{K}_T\big)$, where $\text{diag}(\cdot)$ is the partial diagonalization operation on $\mathcal{X}$:

$$\Sigma_T^{(h)}(\mathbf{x}, t; \mathbf{x}', t') = \Sigma^{(h)}(\mathbf{x}, \mathbf{x}') \cdot K_T(\mathbf{x}, t; \mathbf{x}', t'). \tag{5}$$

The above argument shows that instead of taking care of both the deep learning and continuous-time system, which remains challenging for general architectures, we can convert the problem to finding a suitable temporal kernel. We further point out that when using neural networks, we are parameterizing the hidden representation (feature lift) in the feature space rather than the kernel function in the kernel space. Therefore, to give a consistent characterization, we should also study the feature representation of the temporal kernel and then combine it with the hidden representations of the neural network.

## 3.1 THE RANDOM FEATURE REPRESENTATION FOR TEMPORAL KERNEL

We start by considering the simpler case where the temporal kernel is stationary and independent of features: $K_T(t, t') = k(t' - t)$, for some properly scaled positive even function $k(\cdot)$. The classical Bochner's theorem (Bochner, 1948) states that:

$$\psi(t' - t) = \int_{\mathbb{R}} e^{-i(t'-t)\omega} ds(\omega), \quad \text{for some probability density function } s \text{ on } \mathbb{R}, \tag{6}$$

where $s(\cdot)$ is the spectral density function we highlighted in Section 2.2. To compute the integral, we may sample $(\omega_1, \ldots, \omega_m)$ from $s(\omega)$ and use the Monte Carlo method: $\psi(t' - t) \approx \frac{1}{m} \sum_{i=1}^m e^{-i(t'-t)^\intercal \omega}$. Since $e^{-i(t'-t)^\intercal \omega} = \cos\big((t'-t)\omega\big) - i \sin\big((t'-t)\omega\big)$, for the real part, we let:

$$\boldsymbol{\phi}(t) = \frac{1}{\sqrt{m}} \big[ \cos(t\omega_1), \sin(t\omega_1) \ldots, \cos(t\omega_m), \sin(t\omega_m) \big], \tag{7}$$

and it is easy to check that $\psi(t' - t) \approx \langle \boldsymbol{\phi}(t), \boldsymbol{\phi}(t') \rangle$. Since $\boldsymbol{\phi}(t)$ is constructed from random samples, we refer to it as the *random feature representation* of $K_T$. Random feature has been extensive studied in the kernel machine literature, however, we propose a novel application for random features to parametrize an unknown kernel function. A straightforward idea is to parametrize the spectral density function $s(\omega)$, whose pivotal role has been highlighted in Section 2.2 and Example 1.

Suppose $\boldsymbol{\theta}_T$ is the distribution parameters for $s(\omega)$. Then $\boldsymbol{\phi}_{\boldsymbol{\theta}_T}(t)$ is also (implicitly) parameterized by $\boldsymbol{\theta}_T$ through the samples $\{\omega(\boldsymbol{\theta}_T)_i\}_{i=1}^m$ from $s(\omega)$. The idea resembles the reparameterization trick for training variational objectives (Kingma & Welling, 2013), which we formalize in the next section. For now, it remains unknown if we can also obtain the random feature representation for non-stationary kernels where Bochner's theorem is not applicable. Note that for a general temporal kernel $K_T(\mathbf{x}, t; \mathbf{x}', t')$, in practice, it is not reasonable to assume stationarity specially for the feature domain. In Proposition 1, we provide a rigorous result that generalizes the random feature representation for nonstationary kernels with convergence guarantee.

**Proposition 1.** *For any (scaled) continuous non-stationary PDS kernel $K_T$ on $\mathcal{X} \times \mathcal{T}$, there exists a joint probability measure with spectral density function $s(\boldsymbol{\omega}_1, \boldsymbol{\omega}_2)$, such that $K_T\big((\mathbf{x}, t), (\mathbf{x}', t')\big) = \mathbb{E}_{s(\boldsymbol{\omega}_1, \boldsymbol{\omega}_2)}\big[\boldsymbol{\phi}(\mathbf{x}, t)^\intercal \boldsymbol{\phi}(\mathbf{x}', t')\big]$ where $\boldsymbol{\phi}(\mathbf{x}, t)$ is given by:*

$$\frac{1}{2\sqrt{m}} \big[ \ldots, \cos\big([\mathbf{x}, t]^\intercal \boldsymbol{\omega}_{1,i}\big) + \cos\big([\mathbf{x}, t]^\intercal \boldsymbol{\omega}_{2,i}\big), \sin\big([\mathbf{x}, t]^\intercal \boldsymbol{\omega}_{1,i}\big) + \sin\big([\mathbf{x}, t]^\intercal \boldsymbol{\omega}_{2,i}\big) \ldots \big], \tag{8}$$

*Here, $\{(\boldsymbol{\omega}_{1,i}, \boldsymbol{\omega}_{2,i})\}_{i=1}^m$ are the $m$ samples from $s(\boldsymbol{\omega}_1, \boldsymbol{\omega}_2)$. When the sample size $m \geq \frac{8(d+1)}{\varepsilon^2} \log\Big(C(d)\big(l^2 t_{\max}^2 \sigma_p / \varepsilon\big)^{\frac{2d+2}{d+3}} / \delta\Big)$, with probability at least $1 - \delta$, for any $\varepsilon > 0$,*

$$\sup_{(\mathbf{x}, t), (\mathbf{x}', t')} \Big| K_T\big((\mathbf{x}, t), (\mathbf{x}', t')\big) - \boldsymbol{\phi}(\mathbf{x}, t)^\intercal \boldsymbol{\phi}(\mathbf{x}', t') \Big| \leq \varepsilon, \tag{9}$$

where $\sigma_p^2$ is the second moment of the spectral density function $s(\boldsymbol{\omega}_1, \boldsymbol{\omega}_2)$ and $C(d)$ is a constant.

We defer the proof to Appendix A.3.2. It is obvious that the new random feature representation in (8) is a generalization of the stationary setting. There are two advantages for using the random feature representation:

- the composition in the kernel space suggested by (5) is equivalent to the computationally efficient operation $\boldsymbol{f}^{(h)}(\mathbf{x}) \circ \phi(\mathbf{x}, t)$ in the feature space (Shawe-Taylor et al., 2004);
- we preserve a similar Gaussian process behavior and the neural tangent kernel results that we discussed in Section 2.1, and we defer the discussion and proof to Appendix A.3.3.

In the forward-passing computations, we simply replace the original hidden representation $\boldsymbol{f}^{(h)}(\mathbf{x})$ by the time-aware representation $\boldsymbol{f}^{(h)}(\mathbf{x}) \circ \phi(\mathbf{x}, t)$. Also, the existing methods and results on analyzing neural networks though Gaussian process and NTK, though not emphasized in this paper, can be directly carried out to the temporal setting as well (see Appendix A.3.3).

## 3.2 REPARAMETERIZATION WITH THE SPECTRAL DENSITY FUNCTION

We now present the gradient computation for the parameters of the spectral distribution using only their samples. We start from the well-studied case where $s(\boldsymbol{\omega})$ is given by a normal distribution $N(\mu, \boldsymbol{\Sigma})$ with parameters $\boldsymbol{\theta}_T = [\mu, \boldsymbol{\Sigma}]$. When computing the gradients of $\boldsymbol{\theta}_T$, instead of sampling from the intractable distribution $s(\boldsymbol{\omega})$, we reparameterize each sample $\boldsymbol{\omega}_i$ via: $\boldsymbol{\Sigma}^{1/2}\epsilon + \mu$, where $\epsilon$ is sampled from a standard multivariate normal distribution. The gradient computations that relied on $\boldsymbol{\omega}$ is now replaced using the easy-to-sample $\epsilon$ and $\boldsymbol{\theta}_T = [\mu, \boldsymbol{\Sigma}]$ now become tractable parameters in the model given $\epsilon$. We illustrate reparameterization in our setting in the following example.

**Example 1.** *Consider a single-dimension homogeneous linear model: $f(\theta, x) = f^{(0)}(x) = \theta x$. Without loss of generality, we use only a single sample $\omega_1$ from the $s(\omega)$ which corresponds to the feature-independent temporal kernel $k_{\boldsymbol{\theta}}(t, t')$. Again, we assume $s(\omega) \sim N(\mu, \sigma)$.*

*Then the time-aware hidden representation for this layer for datapoint $(x_1, t_1)$ is given by:*

$$f^{(0)}_{\theta, \mu, \sigma}(x_1, t_1) = 1/\sqrt{2}[\theta x_1 \cos(t_1 \omega_1), \theta x_1 \sin(t_1 \omega_1)], \ \omega_1 \sim N(\mu, \sigma).$$

*Using the reparameterization, given a sample $\epsilon_1$ from the standard normal distribution, we have:*

$$f^{(0)}_{\theta, \mu, \sigma}(x_1, t_1) = 1/\sqrt{2}[\theta x_1 \cos(t_1(\sigma^{1/2}\epsilon_1 + \mu)), \theta x_1 \sin(t_1(\sigma^{1/2}\epsilon_1 + \mu))], \quad (10)$$

*so the gradients with respect to all the parameters $(\theta, \mu, \sigma)$ can be computed in the usual way.*

Despite the computation advantage, the spectral density is now learnt from the data instead of being given so the convergence result in Proposition 1 does not provide sample-consistency guarantee. In practice, we may also misspecify the spectral distribution and bring extra intractable factors.

To provide practical guarantees, we first introduce several notations: let $\mathbf{K}_T(S)$ be the temporal kernel represented by random features such that $\mathbf{K}_T(S) = \mathbb{E}[\phi^\intercal \phi]$, where the expectation is taken with respect to the data distribution and the random feature vector $\phi$ has its samples $\{\omega_i\}_{i=1}^m$ drawn from the spectral distribution $S$. Without abuse of notation, we use $\phi \sim S$ to denote the dependency of the random feature vector $\phi$ on the spectral distribution $S$ provided in (8). Given a neural network kernel $\boldsymbol{\Sigma}^{(h)}$, the *neural temporal kernel* is then denoted by: $\boldsymbol{\Sigma}_T^{(h)}(S) = \boldsymbol{\Sigma}^{(h)} \otimes \mathbf{K}_T(S)$. So the sample version of $\boldsymbol{\Sigma}_T^{(h)}(S)$ for the dataset $\{(\mathbf{x}_i, t_i)\}_{i=1}^n$ is given by:

$$\hat{\boldsymbol{\Sigma}}_T^{(h)}(S) = \frac{1}{n(n-1)} \sum_{i \neq j} \Sigma^{(h)}(\mathbf{x}_i, \mathbf{x}_j) \phi(\mathbf{x}_i, t_i)^\intercal \phi(\mathbf{x}_j, t_j), \ \phi \sim S. \quad (11)$$

If the spectral distribution $S$ is fixed and given, then using standard techniques and Theorem 1 it is straightforward to show $\lim_{n \to \infty} \hat{\boldsymbol{\Sigma}}_T^{(h)}(S) \to \mathbb{E}[\hat{\boldsymbol{\Sigma}}_T^{(h)}(S)]$ so the proposed learning schema is sample-consistent.

In our case, the spectral distribution is learnt from the data, so we need some restrictions on the spectral distribution in order to obtain any consistency guarantee. The intuition is that if $S_{\boldsymbol{\theta}_T}$ does not diverge from the true $S$, e.g. $d(S_{\boldsymbol{\theta}_T} \| S) \leq \delta$ for some divergence measure, the guarantee on $S$ can transfer to $S_{\boldsymbol{\theta}_T}$ with the rate only suffers a discount that does not depend on $n$.

**Theorem 1.** *Consider the f-divergence such that $d(S_{\boldsymbol{\theta}_T} \| S) = \int c(dS_{\boldsymbol{\theta}_T}/dS)dS$, with the generator function $c(x) = x^k - 1$ for any $k > 0$. Given the neural network kernel $\boldsymbol{\Sigma}^h$, let $M = \left\| \boldsymbol{\Sigma}^h \right\|_\infty$, then*

$$Pr\Big( \sup_{d(S_{\boldsymbol{\theta}_T}\|S)\leq\delta} \big| \hat{\boldsymbol{\Sigma}}_T^{(h)}(S_{\boldsymbol{\theta}_T}) \to \mathbb{E}\big[ \hat{\boldsymbol{\Sigma}}_T^{(h)}(S_{\boldsymbol{\theta}_T}) \big| \geq \varepsilon \Big) \leq \sqrt{2}\exp\Big( \frac{-n\varepsilon^2}{64\max\{4,M\}(\delta+1)} \Big) + C(\varepsilon),$$

(12)

*where $C(\varepsilon) \propto \Big( \frac{2l^2 t_{\max}^2 \sigma_{S_{\boldsymbol{\theta}_T}}}{\epsilon/\max\{4,M\}} \Big)^{\frac{2d+2}{d+3}} \exp\Big( -\frac{d_h \epsilon^2}{32\max\{16,M^2\}(d+3)} \Big)$ that does not depend on $\delta$.*

The proof is provided in Appendix A.3.4. The key takeaway from (12) is that as long as the divergence between the learnt $S_{\boldsymbol{\theta}_T}$ and the true spectral distribution is bounded, we still achieve sample consistency. Therefore, instead of specifying a distribution family which is more likely to suffer from misspecification, we are motivated to employ some universal distribution approximator such as the *invertible neural network* (INN) (Ardizzone et al., 2018). INN consists of a series of invertible operations that transform samples from a known auxiliary distribution (such as normal distribution) to arbitrarily complex distributions. The Jacobian that characterize the changes of distributions are made invertible by the INN, so the gradient flow is computationally tractable similar to the case in Example 1. We defer the detailed discussions to Appendix A.3.5.

**Remark 1.** *It is clear at this point that the temporal kernel approach applies to all the neural networks who have a Gaussian process behavior with a valid neural network kernel, which includes the major architectures such as CNN, RNN and attention mechanism (Yang, 2019).*

For implementation, at each forward and backward computation, we first sample from the auxiliary distribution to construct the random feature representation $\phi$ using reparameterization, and then compose it with the selected hidden layer $\boldsymbol{f}^{(h)}$ such as in (10). We illustrate the computation architecture in Figure 2, where we adapt the vanilla RNN to the proposed framework. In Algorithm 1, we provide the detailed forward and backward computations, using the L-layer FFN from the previous sections as an example.

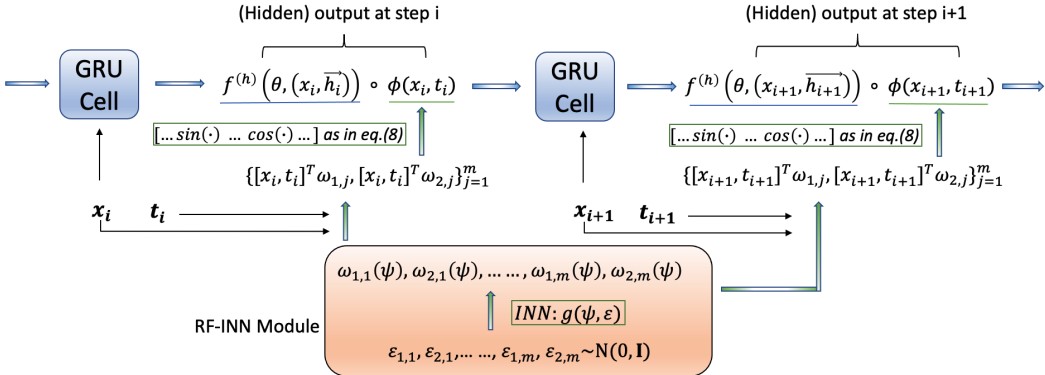

Figure 2: The computation architecture of the proposed method using RNN as an example (following Figure 1b). Here, we use $f^{(h)}(\boldsymbol{\theta}, \cdot)$ to denote the standard GRU cell, and use $g(\boldsymbol{\psi}, \cdot)$ to denote the invertible neural network. At the $i^{\text{th}}$ step, the GRU cell takes the feature vector $\mathbf{x}_i$, and the hidden state of the previous steps $\vec{\mathbf{h}}_i$, as input. The RF-INN module is called once for each batch.

## 4  RELATED WORK

The earliest work that discuss training continuous-time neural network dates back to LeCun et al. (1988); Pearlmutter (1995), but no feasible solution was proposed at that time. The proposed approach relates to several fields that are under active research.

**ODE and neural network.** Certain neural architecture such as the residual network has been interpreted as approximate ODE solvers (Lu et al., 2018). More direct approaches have been proposed to learn differential equations from data (Raissi & Karniadakis, 2018; Raissi et al., 2018a;

Long et al., 2018), and significant efforts have been spent on developing solvers that combine ODE and the back-propagation framework (Farrell et al., 2013; Carpenter et al., 2015; Chen et al., 2018). The closest literature to our work is from Raissi et al. (2018b) who design numerical Gaussian process resulting from temporal discretization of time-dependent partial differential equations.

**Random feature and kernel machine learning.** In supervised learning, the kernel trick provides a powerful tool to characterize non-linear data representations (Shawe-Taylor et al., 2004), but the computation complexity is overwhelming for large dataset. The random (Fourier) feature approach proposed by Rahimi & Recht (2008) provides substantial computation benefits. The existing literature on analyzing the random feature approach all assume the kernel function is fixed and stationary (Yang et al., 2012; Sutherland & Schneider, 2015; Sriperumbudur & Szabó, 2015; Avron et al., 2017).

**Reparameterization and INN.** Computing the gradient for intractable objectives using samples from auxiliary distribution dates back to the policy gradient method in reinforcement learning (Sutton et al., 2000). In recent years, the approach gains popularity for training generative models (Kingma & Welling, 2013), other variational objectives (Blei et al., 2017) and Bayesian neural networks (Snoek et al., 2015). INN are often employed to parameterize the *normalizing flow* that transforms a simple distribution into a complex one by applying a sequence of invertible transformation functions (Dinh et al., 2014; Ardizzone et al., 2018; Kingma & Dhariwal, 2018; Dinh et al., 2016).

Our approach characterizes the continuous-time ODE via the lens of kernel. It complements the existing neural ODE methods which are often restricted to specific architectures, relying on ODE solvers and lacking theoretical understandings. We also propose a novel deep kernel learning approach by parameterizing the spectral distribution under random feature representation, which is conceptually different from using temporal kernel for time-series classification (Li & Marlin, 2015). Our work is a extension of Xu et al. (2019; 2020), which study the case for self-attention mechanism.

---

**Algorithm 1:** Forward pass and parameter update, using the L-layer FFN as an example.

---

**Input:** The FFN $f(\boldsymbol{\theta}, \cdot) = \{f^{(1)}(\boldsymbol{\theta}, \cdot), \ldots, f^{(L)}(\boldsymbol{\theta}, \cdot)\}$; the invertible neural network $g(\boldsymbol{\psi}, \cdot)$; the selected hidden layer $h$; the loss $\ell_i$ associated with each input $(\mathbf{x}_i, t_i)$; the auxilliary distribution $P$.

**for** *each mini-batch* **do**

    Sample $\left\{\boldsymbol{\epsilon}_{1,j}, \boldsymbol{\epsilon}_{2,j}\right\}_{j=1}^{m}$ from the auxilliary distribution $P$;

    Compute the *reparameterized* samples $\boldsymbol{\omega}$ using the INN $g(\boldsymbol{\psi}, \cdot)$, e.g. $\boldsymbol{\omega}_{1,j}(\boldsymbol{\psi}) := g(\boldsymbol{\psi}, \boldsymbol{\epsilon}_{1,j})$;

    **for** *sample $i$ in the batch* **do**

        Construct the random feature representation $\boldsymbol{\phi}_{\boldsymbol{\psi}}(\mathbf{x}_i, t_i)$ using the *reparameterized* samples (so $\boldsymbol{\phi}$ is now explicitly parameterized by $\boldsymbol{\psi}$) according to eq. (8);

        **Forward pass:** get $f^{(h)}(\boldsymbol{\theta}, \mathbf{x}_i)$, let $f^{(h)}\big((\boldsymbol{\theta}, \boldsymbol{\psi}), \mathbf{x}_i, t_i\big) := f^{(h)}(\boldsymbol{\theta}, \mathbf{x}_i) \circ \boldsymbol{\phi}_{\boldsymbol{\psi}}(\mathbf{x}_i, t_i)$, then pass it to the following feedforward layers to obtain the final output $\hat{y}_i$ ;

        **Gradient computation**: compute the gradients $\nabla_{\boldsymbol{\theta}}\ell_i(\hat{y}_i)\big|_{\boldsymbol{\epsilon}}$, $\nabla_{\boldsymbol{\psi}}\ell_i(\hat{y}_i)\big|_{\boldsymbol{\epsilon}}$ for the FFN and INN respectively, conditioned on the samples from the auxiliary distribution;

    **end**

    Update the parameters using the selected optimizer in a standard batch-wise fashion.

**end**

---

It is straightforward from Figure 2 and Algorithm 1 that the proposed approach serves as a plug-in module and do not modify the original network structures of the RNN and FFN.

## 5 EXPERIMENTS AND RESULTS

We focus on revealing the two major advantages of the proposed temporal kernel approach:

- the temporal kernel approach consistently improves the performance of deep learning models, both for the general architectures such as RNN, CausalCNN and attention mechanism as well as the domain-specific architectures, in the presence of continuous-time information;

- the improvement is not at the cost of computation efficiency and stability, and we outperform the alternative approaches who also applies to general deep learning models.

We point out that the neural point process and the ODE neural networks have only been shown to work for certain model architectures so we are unable to compare with them for all the settings.

**Time series prediction with standard neural networks (real-data and simulation)**

We conduct time series prediction task using the vanilla RNN, CausalCNN and self-attention mechanism with our temporal kernel approach (Figure A.1). We choose the classical **Jena weather data** for temperature prediction, and the **Wikipedia traffic data** to predict the number of visits of Wikipedia pages. Both datasets have vectorized features and are regular-sampled. To illustrate the advantage of leveraging the temporal information compared with using only sequential information, we first conduct the ordinary next-step prediction on the regular observations, which we refer to as **Case1**. To fully illustrate our capability of handling the irregular continuous-time information, we consider the two **simulation setting** that generate irregular continuous-time sequences for prediction:

**Case2**. we sample irregularly from the history, i.e. $\mathbf{x}_{t_1}, \ldots, \mathbf{x}_{t_q}$, $q \leq k$, to predict $\mathbf{x}_{t_{k+1}}$;

**Case3**. we use the full history to predict a dynamic future point, i.e. $\mathbf{x}_{t_{k+q}}$ for a random $q$.

We provide the complete data description, preprocessing, and implementation in Appendix B. We use the following two widely-adopted time-aware modifications for neural networks (denote by **NN**) as baselines, as well as the classical vectorized autoregression model (**VAR**).
**NN+time**: we directly concatenate the timespan, e.g. $t_j - t_i$, to the feature vector. **NN+trigo**: we concatenate the learnable sine and cosine features, e.g. $[\sin(\pi_1 t), \ldots, \sin(\pi_k t)]$, to the feature vector, where $\{\pi_i\}_{i=1}^k$ are free model parameters. We denote our temporal kernel approach by **T-NN**.
From Figure 3, we see that the temporal kernel outperforms the baselines in all cases when the time series is irregularly sampled (**Case2** and **Case3**), suggesting the effectiveness of the temporal kernel approach in capturing and utilizing the continuous-time signals. Even for the regular **Case1** reported in Table A.1, the temporal kernel approach gives the best results, which again emphasizes the advantage of directly characterize the temporal information over discretization. We also show in the ablation studies (Appendix B.5) that INN is necessary for achieving superior performance compared with specifying a distribution family. To demonstrate the stability and robustness, we provide sensitivity analysis in Appendix B.6 for model selection and INN structures.

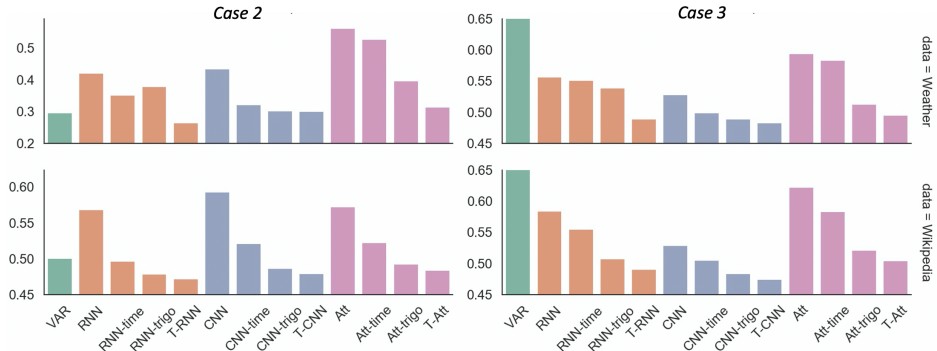

Figure 3: The *mean absolute error* on testing data for the standard neural networks: RNN, CausalCNN (denoted by CNN) and self-attention (denoted by Att), for the temporal kernel approach and the baselines methods in **Case2** and **Case3**. The numerical results are averaged over five repetitions.

**Temporal sequence learning with complex domain models**

Now we study the performance of our temporal kernel approach for the sequential recommendation task with more complicated domain-specific two-tower architectures (Appendix B.2). Temporal information is known to be critical for understanding customer intentions, so we choose the two public e-commerce dataset from **Alibaba** and **Walmart.com**, and examine the next-purchase recommendation. To illustrate our flexibility, we select the GRU-based, CNN-based and attention-based recommendation models from the recommender system domain (Hidasi et al., 2015; Li et al., 2017a) and equip them with the temporal kernel. The detailed settings, ablation studies and sensitivity analysis are all in Appendix B. The results are shown in Table A.2. We observe that the temporal kernel approach brings various degrees of improvements to the recommendation models by characterizing

the continuous-time information. The positives results from the recommendation task also suggests the potential of our approach for making impact in boarder domains.

## 6 DISCUSSION

In this paper, we discuss the insufficiency of existing work on characterizing continuous-time data with deep learning models and describe a principled temporal kernel approach that expands neural networks to characterize continuous-time data. The proposed learning approach has strong theoretical guarantees, and can be easily adapted to a broad range of applications such as deep spatial-temporal modelling, outlier and burst detection, and generative modelling for time series data.

**Scope and limitation**. Although the temporal kernel approach is motivated by the limiting-width Gaussian behavior of neural networks, in practice, it suffices to use regular widths as we did in our experiments (see Appendix B.2 for the configurations). Therefore, there are still gaps between our theoretical understandings and the observed empirical performance, which require more dedicated analysis. One possible direction is to apply the techniques in Daniely et al. (2016) to characterize the dual kernel view of finite-width neural networks. The technical detail, however, will be more involved. It is also arguably true that we build the connection between the temporal kernel view and continuous-time system in an indirect fashion, compared with the ODE neural networks. However, our approach is fully compatible with the deep learning subroutines while the end-to-end ODE neural networks require substantial modifications to the modelling and implementation. Nevertheless, ODE neural networks are (in theory) capable of modelling more complex systems where the continuous-time setting is a special case. Our work, on the other hand, is dedicated to the temporal setting.

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

## A   APPENDIX

We provide the omitted proofs, detailed discussions, extensions and complete numerical results.

### A.1   NUMERICAL RESULTS FOR SECTION 5

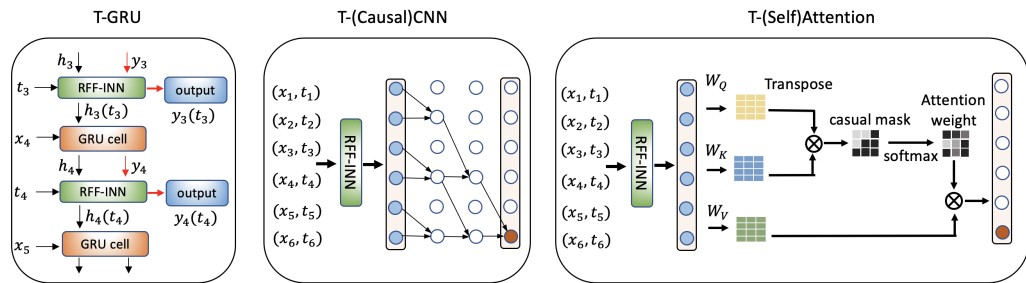

Figure A.1: Visual illustrations on how we equip the standard neural architectures with the temporal kernel using the random Fourier feature with invertible neural network (the **RFF-INN** blocks).

| Case 1 | | |
|---|---|---|
| | *Weather* | *Wikipedia* |
| VAR | 0.2643 | 2.31 |
| RNN | 0.2487/.002 | 0.5142/.003 |
| RNN-time | 0.2629/.001 | 0.4698/.004 |
| RNN-trigo | 0.2526/.003 | 0.4542/.004 |
| **T-RNN** | ***0.2386/.002*** | *0.4330/.002* |
| CNN | 0.3103/.002 | 0.4998/.003 |
| CNN-time | 0.2933/.003 | 0.4852/.001 |
| CNN-trigo | 0.2684/.004 | 0.4556/.002 |
| **T-CNN** | *0.2662/.003* | *0.4399/.003* |
| Attention | 0.4052/.003 | 0.4795/.003 |
| Attention-time | 0.4298/.003 | 0.4809/.002 |
| Attention-trigo | 0.2887/.002 | 0.4445/.003 |
| **T-Attention** | *0.2674/.004* | ***0.4226/.002*** |

Table A.1: *Mean absolute error* for time series prediction of the regular scenario of **Case 1**. We underline the best results for each neural architecture, and the overall best results are highlighted in bold-font. The reported results are averaged over five repetitions, with the standard errors provided.

| | *Alibaba* | | *Walmart* | |
|---|---|---|---|---|
| Metric | Accuracy | DCG | Accuracy | DCG |
| GRU-Rec | 77.81/.21 | 47.12/.11 | 18.09/.13 | 3.44/.21 |
| GRU-Rec-time | 77.70/.24 | 46.21/.13 | 17.66/.16 | 3.29/.23 |
| GRU-Rec-trigo | 78.95/.19 | 49.01/.11 | 21.54/.13 | 6.67/.18 |
| **T-GRU-Rec** | ***79.47/.35*** | ***49.82/.40*** | *23.41/.11* | *8.44/.21* |
| CNN-Rec | 74.89/.33 | 43.91/.22 | 15.98/.18 | 1.97/.19 |
| CNN-Rec-time | 74.85/.31 | 43.88/.21 | 15.95/.18 | 1.96/.17 |
| CNN-Rec-trigo | 75.97/.21 | 45.86/.23 | 17.74/.17 | 3.80/.15 |
| **T-CNN-Rec** | *76.45/.16* | *46.55/.38* | *18.59/.33* | *4.56/.31* |
| ATTN-Rec | 51.82/.44 | 30.41/.52 | 20.41/.38 | 7.52/.18 |
| ATTN-Rec-time | 51.84/.43 | 30.45/.50 | 20.43/.36 | 7.54/.19 |
| ATTN-Rec-trigo | 53.05/.30 | 33.10/.29 | 24.49/.15 | 8.93/.13 |
| **T-ATTN-Rec** | *53.49/.31* | *33.58/.30* | ***25.51/.17*** | ***9.22/.15*** |

Table A.2: *Accuracy* and *discounted cumulative gain (DCG)* for the domain-specific models on the temporal recommendation tasks. See Appendix B for detail. We underline the best results for each neural architecture, and the overall best results are highlighted in bold-font.

## A.2 SUPPLEMENTARY MATERIAL FOR SECTION 2

We discuss the detailed background for the Gaussian process behavior of neural network and the training trajectory under neural tangent kernel, as well as the proof for Claim 1.

### A.2.1 GAUSSIAN PROCESS BEHAVIOR AND NEURAL TANGENT KERNEL FOR DEEP LEARNING MODELS

The *Gaussian process (GP)* view of neural networks at random initialization was originally discussed in (Neal, 2012). Recently, CNN and other standard neural architectures have all been recognized as functions drawn from GP in the limit of infinite network width (Novak et al., 2018; Yang, 2019). When trained by gradient descent under infinitesimal step schedule, the gradient flow of the standard neural architectures can be described by the notion of *Neural Tangent Kernel (NTK)* whose asymptotic behavior under infinite network width is known (Jacot et al., 2018). The discovery of NTK has led to several papers studying the training and generalization properties of neural networks (Allen-Zhu et al., 2019; Arora et al., 2019a;b).

For a L-layer FNN $f(\boldsymbol{\theta}, \mathbf{x}) = f^{(L)}$ with hidden dimensions $\{d_h\}_{h=1}^L$ and recursively defined via:

$$f^{(L)} = \mathbf{W}^{(L)} \boldsymbol{f}^{(L)}(\mathbf{x}) + b^{(L)}, \quad \boldsymbol{f}^{(h)}(\mathbf{x}) = \frac{1}{\sqrt{d_h}} \mathbf{W}^{(h)} \sigma\big(\boldsymbol{f}^{(h-1)}(\mathbf{x})\big) + \mathbf{b}^{(h)}, \quad \boldsymbol{f}^{(0)}(\mathbf{x}) = \mathbf{x},$$

(A.1)

for $h = 1, 2, \ldots, L - 1$, where $\sigma(.)$ is the activation function and the layer weights $\mathbf{W}^{(L)} \in \mathbb{R}^{d_{L-1}}$, $\mathbf{W}^{(h)} \in \mathbb{R}^{d_{h-1} \times d_h}$ and intercepts are initialized by sampling independently from $\mathcal{N}(0, 1)$ (without loss of generality). As $d_1, \ldots, d_L \to \infty$, $\boldsymbol{f}^{(h)}$ tend in law to i.i.d Gaussian processes with covariance $\Sigma^h$ defined recursively as shown by Neal (2012):

$$\Sigma^{(1)}(\mathbf{x}, \mathbf{x}') = \frac{1}{h_1} \mathbf{x}^\intercal \mathbf{x}' + 1, \quad \Sigma^{(h)}(\mathbf{x}, \mathbf{x}') = \mathbb{E}_{f \sim \mathcal{N}(0, \Sigma^{(h-1)})} \big[\sigma\big(f(\mathbf{x})\big) \sigma\big(f(\mathbf{x}')\big)\big] + 1. \quad \text{(A.2)}$$

We also refer to $\Sigma^{(h)}$ as the *neural network kernel* to distinguish from the other kernel notions. Given a training dataset $\{\mathbf{x}_i, y_i\}_{i=1}^n$, let $\boldsymbol{f}\big(\boldsymbol{\theta}(s)\big) = \big(f(\boldsymbol{\theta}(s), \mathbf{x}_1), \ldots, f(\boldsymbol{\theta}(s), \mathbf{x}_n)\big)$ be the network outputs at the $s^{th}$ training step and $\mathbf{y} = (y_1, \ldots, y_n)$.

When training the network by minimizing the *squared loss* $\ell(\boldsymbol{\theta})$ with infinitesimal learning rate, i.e. $\frac{d\boldsymbol{\theta}(s)}{ds} = -\nabla\ell(\boldsymbol{\theta}(s))$, the network outputs at training step $s$ follows the evolution (Jacot et al., 2018):

$$\frac{d\boldsymbol{f}\big(\boldsymbol{\theta}(s)\big)}{ds} = -\boldsymbol{\Theta}(s) \times (\boldsymbol{f}\big(\boldsymbol{\theta}(s)\big) - \mathbf{y}), \quad \big[\boldsymbol{\Theta}(s)\big]_{ij} = \Big\langle \frac{\partial f(\boldsymbol{\theta}(s), x_i)}{\partial \boldsymbol{\theta}}, \frac{\partial f(\boldsymbol{\theta}(s), x_j)}{\partial \boldsymbol{\theta}} \Big\rangle. \quad \text{(A.3)}$$

The above $\boldsymbol{\Theta}(s)$ is referred to as the NTK, and recent results shows that when the network widths go to infinity (or sufficiently large), $\boldsymbol{\Theta}(s)$ converges to a fixed $\boldsymbol{\Theta}_0$ almost surely (or with high probability).

For a standard L-layer FFN, the NTK $\boldsymbol{\Theta}_0 = \boldsymbol{\Theta}_0^{(L)}$ for parameters $\{\mathbf{W}^{(h)}, \mathbf{b}^{(h)}\}$ on the $h^{\text{th}}$-layer can also be computed recursively:

$$\boldsymbol{\Theta}_0^{(h)}(\mathbf{x}_i, \mathbf{x}_j) = \Sigma^{(h)}(\mathbf{x}_i, \mathbf{x}_j)$$
$$\dot{\Sigma}^{(k)}(\mathbf{x}_i, \mathbf{x}_j) = \mathbb{E}_{f \sim \mathcal{N}(0, \Sigma^{(k-1)})} \big[\dot{\sigma}(f(\mathbf{x}_i)) \dot{\sigma}(f(\mathbf{x}_j))\big], \quad \text{(A.4)}$$
$$\text{and } \boldsymbol{\Theta}_0^{(k)}(\mathbf{x}_i, \mathbf{x}_j) = \boldsymbol{\Theta}_0^{(k-1)}(\mathbf{x}_i, \mathbf{x}_j) \dot{\Sigma}^{(k)}(\mathbf{x}_i, \mathbf{x}_j) + \Sigma^{(k)}(\mathbf{x}_i, \mathbf{x}_j), \quad k = h + 1, \ldots, L.$$

A number of optimization and generalization properties of neural networks can be studied using NTK, which we refer the interested readers to (Lee et al., 2019; Allen-Zhu et al., 2019; Arora et al., 2019a;b). We also point out that the above GP and NTK constructions can be carried out on all standard neural architectures including CNN, RNN and the attention mechanism (Yang, 2019).

### A.2.2 PROOF FOR CLAIM 1

In this part, we denote the continuous-time system by $X(t)$ in order to introduce the full set notations that are needed for our proof, where the length of the discretized interval is explicitly considered. Note that we especially construct the example in Section 2 so that the derivations are not too cumbersome. However, the techniques that we use here can be extended to prove the more complicated settings.

*Proof.* Consider $X(t)$ to be the second-order continuous-time autoregressive process with covariance function $k(t)$ and spectral density function (SDF) $s(\omega)$ such that $s(\omega) = \int_{-\infty}^{\infty} \exp(-i\omega t)k(t)dt$. The covariance function of the discretization $X_a[n] = X(na)$ with any fixed interval $a > 0$ is then given by $k_a[n] = k(na)$. According to standard results in time series, the SDF of $X(t)$ is given by in the form of:

$$s(\omega) = \frac{a_1}{\omega^2 + b_1^2} + \frac{a_2}{\omega^2 + b_2^2}, \quad a_1 + a_2 = 0, \quad a_1 b_2^2 + a_2 b_1^2 \neq 0. \tag{A.5}$$

We assume without loss of generality that $b_1, b_2$ are positive numbers. Note that the kernel function for $X_a[i]$ can also be given by

$$
\begin{aligned}
k_a[n] &= \int_{-\infty}^{\infty} \exp(ian\omega)s(\omega)d\omega \\
&= \frac{1}{a} \sum_{k=-\infty}^{\infty} \int_{(2k-1)\pi}^{(2k+1)\pi} \exp(in\omega)s(\omega/a)d\omega \\
&= \frac{1}{a} \int_{-\infty}^{\infty} \exp(in\omega) \sum_{k=-\infty}^{\infty} s\left(\frac{\omega + 2k\pi}{a}\right)d\omega,
\end{aligned}
\tag{A.6}
$$

which suggests that the SDF for the discretization $X_a[n]$ can be given by:

$$
\begin{aligned}
s_a(\omega) &= \frac{1}{a} \sum_{k=-\infty}^{\infty} s\left(\frac{\omega + 2k\pi}{h}\right) \\
&= \frac{a_1}{2} \left(\frac{e^{2ab_1} - 1}{b_1|e^{ab_1} - e^{i\omega}|^2} - \frac{e^{2ab_2} - 1}{b_1|e^{ab_2} - e^{i\omega}|^2}\right) \\
&= \frac{a_1(d_1 - 2d_2\cos(\omega))}{2b_1b_2|(e^{ab_1} - e^{i\omega})(e^{ab_2} - e^{i\omega})|^2},
\end{aligned}
\tag{A.7}
$$

where $d_2 = b_2 e^{ab_2}(e^{2ab_2} - 1) - b_1 e^{ab_1}(e^{2ab_1} - 1)$. By the definition of discrete-time auto-regressive process, $X_a[n]$ is a second-order AR process only if $d_2 = 0$, which happens if and only if: $b_2/b_1 = (e^{ab_2} - e^{-ab_2})/(e^{ab_1} - e^{-ab_1})$. However, the function $g(x) = \exp(ax) - \exp(-ax)$ is concave on $[0, \infty)$ (since the time interval $a > 0$) and $g(0) = 0$, the-above equality hold if $b_1 = b_2$. However, this contradicts with (A.5), since $a_1 + a_2 = 0$ and $a_1 b_2^2 + a_2 b_1^2 \neq 0$ suggests $a_1(b_1 - b_2)^2 \neq 0$. Hence, $X_a[n]$ cannot be a second-order discrete-time auto-regressive process. $\square$

## A.3 SUPPLEMENTARY MATERIAL FOR SECTION 3

We first present the related discussions and proof for Claim 2 on the connection between continuous-time system and temporal kernel. Then we prove the convergence result in Theorem 1 regarding the random feature representation for non-stationary kernel. In the sequel, we show the new results for the Gaussian process behavior and neural tangent kernel under random feature representations, and discuss the potential usage of our results. Finally, we prove the sample-consistency result when the spectral distribution is misspecified.

### A.3.1 PROOF AND DISCUSSIONS FOR CLAIM 2

*Proof.* Recall that we study the dynamic system given by:

$$a_n(\mathbf{x})\frac{d^n f(\mathbf{x}, t)}{dt^n} + \cdots + a_0(\mathbf{x})f(\mathbf{x}, t) = b_m(\mathbf{x})\frac{d^m \epsilon(\mathbf{x}, t)}{dt^m} + \cdots + b_0\epsilon(\mathbf{x}, t), \tag{A.8}$$

where $\epsilon(\mathbf{x}, t = t_0) \sim N(0, \mathbf{\Sigma}^{(h)})$, $\forall t_0 \in \mathcal{T}$. The solution process to the above continuous-time system is also a Gaussian process, since $\epsilon(\mathbf{x}, t)$ is a Gaussian process and the solution of a linear different equation is a linear operation on the input. For the sake of notation, we assume $b_0(\mathbf{x}) = 1$ and $b_1(\mathbf{x}) = 0, \ldots, b_m(\mathbf{x}) = 0$, which does not change the arguments in the proof. We apply the Fourier transformation transformation on both sides and solve the for Fourier transform $\tilde{f}(i\boldsymbol{\omega}_{\mathbf{x}}, i\omega)$:

$$\tilde{f}(i\boldsymbol{\omega}_{\mathbf{x}}, i\omega) = \left(\frac{1}{a_n(\mathbf{x}) \cdot (i\omega)^q + \cdots + a_1(\mathbf{x}) \cdot i\omega + a_0(\mathbf{x})}\right)W(i\omega; \boldsymbol{\omega}_{\mathbf{x}}), \tag{A.9}$$

where $W(i\omega; \boldsymbol{\omega_x})$ is the Fourier transform of $\epsilon(\mathbf{x}, t)$. If we do not make the assumption on $\{b_j(\mathbf{x})\}_{j=1}^m$, they will simply show up on the numeration in the same fashion as $\{a_j(\mathbf{x})\}_{j=1}^n$. Let

$$G_{\boldsymbol{\theta}_T}(i\omega; \mathbf{x}) = a_q(\mathbf{x}) \cdot (i\omega)^q + \cdots + a_1(\mathbf{x}) \cdot i\omega + a_0(\mathbf{x}),$$

and $p(\boldsymbol{\omega_x}) = |W(i\omega; \boldsymbol{\omega_x})|^2$ be the spectral density of the Gaussian process corresponding to $\epsilon$ (its spectral density does not depend on $\omega$ because $\epsilon$ is a Gaussian white noise process on the time dimension). The dependency of $G(\cdot; \cdot)$ on $\boldsymbol{\theta}_T$ is because we defined $\boldsymbol{\theta}_T$ to the underlying parameterization of $\{a_j(\cdot)\}_{j=1}^n$ in the statement of Claim 2. Then the spectral density of the process $f(\mathbf{x}, t)$ is given by

$$p(\omega, \boldsymbol{\omega_x}) = C \cdot p(\boldsymbol{\omega_x})|G_{\boldsymbol{\theta}_T}(i\omega; \mathbf{x})|^2 \propto p(\boldsymbol{\omega_x})p_{\boldsymbol{\theta}_T}(\omega; \mathbf{x}),$$

where $C$ is constant that corresponds to the spectral density of the random Gaussian noise on the time dimension. Notice that the spectral density function obtained this way is regular, since it has the form of $p_{\boldsymbol{\theta}_T}(\omega; \mathbf{x}) = \text{constant}/(\text{polynomial of } \omega^2)$.

Therefore, according to the classical Wiener-Khinchin theorem Brockwell et al. (1991), the covariance function of the solution process is given by the inverse Fourier transform of the spectral density:

$$\begin{aligned}
\psi(\mathbf{x}, t) &= \frac{1}{2\pi} \int p(\omega, \boldsymbol{\omega_x}) \exp\left(i[\omega, \boldsymbol{\omega_x}]^\mathsf{T}[t, \mathbf{x}]\right) d(\omega, \boldsymbol{\omega_x}) \\
&\propto \int p_{\boldsymbol{\theta}_T}(\omega; \mathbf{x}) \exp(i\omega t) d\omega \cdot \int p(\boldsymbol{\omega_x}) \exp(i\boldsymbol{\omega_x^\mathsf{T}}\mathbf{x}) d\boldsymbol{\omega_x} \\
&\propto K_{\boldsymbol{\theta}_T}\left((\mathbf{x}, t), (\mathbf{x}, t)\right) \cdot \Sigma^{(h)}(\mathbf{x}, \mathbf{x}).
\end{aligned} \tag{A.10}$$

And therefore we reach the conclusion in Claim 2 by taking $\Sigma_T^{(h)}(\mathbf{x}, t; \mathbf{x}', t') = \psi(\mathbf{x} - \mathbf{x}', t - t')$. $\quad\square$

The inverse statement of Claim 2 may not be always true, since not all the neural-temporal kernel can find a exact corresponding continuous-time system in the form of (A.8). However, we may construct the continuous-time system that approximates the kernel (arbitrarily well) in the following way, using the polynomial approximation tools such as the Taylor expansion.

For a neural-temporal kernel $\Sigma_T^{(h)}$, we first compute it Fourier transform to obtain the spectral density $p(\boldsymbol{\omega_x}, \omega)$. Note that $p(\boldsymbol{\omega_x}, \omega)$ should be a rational function in the form of (polynomial in $\omega^2$)/(polynomial in $\omega^2$), or otherwise it does not have stable spectral factorization that leads to a linear dynamic system. To achieve the goal, we can always apply Taylor expansion or Pade approximants that recovers the $p(\boldsymbol{\omega_x}, \omega)$ arbitrarily well.

Then we conduct a spectral factorization on $p(\boldsymbol{\omega_x}, \omega)$ to find $G(i\boldsymbol{\omega_x}, i\omega_t)$ and $p(\boldsymbol{\omega_x})$ such that $p(\boldsymbol{\omega_x}, \omega) = G(i\boldsymbol{\omega_x}, i\omega_t)p(\boldsymbol{\omega_x})G(-i\boldsymbol{\omega_x}, -i\omega_t)$. Since $p(\boldsymbol{\omega_x}, \omega)$ is now in a rational function form of $\omega^2$, we can find $G(i\boldsymbol{\omega_x}, i\omega_t)$ as:

$$\frac{b_k(i\boldsymbol{\omega_x}) \cdot (i\omega)^k + \cdots + b_1(i\boldsymbol{\omega_x}) \cdot (i\omega) + b_0(i\boldsymbol{\omega_x})}{a_q(i\boldsymbol{\omega_x}) \cdot (i\omega)^q + \cdots + a_1(i\boldsymbol{\omega_x}) \cdot (i\omega) + a_0(i\boldsymbol{\omega_x})}.$$

Let $\alpha_j(\mathbf{x})$ and $\beta_j(\mathbf{x})$ be the pseudo-differential operators of $a_j(i\boldsymbol{\omega_x})$ and $b_j(i\boldsymbol{\omega_x})$ defined in terms of their inverse Fourier transforms (Shubin, 1987), then the corresponding continuous-time system is given by:

$$\alpha_q(\mathbf{x})\frac{\mathrm{d}^q f(\mathbf{x}, t)}{\mathrm{d}t^q} + \cdots + \alpha_0(\mathbf{x})f(\mathbf{x}, t) = \beta_k(\mathbf{x})\frac{\mathrm{d}^k \epsilon(t)}{\mathrm{d}t^k} + \cdots + \beta_0(\mathbf{x})\epsilon(t). \tag{A.11}$$

For a concrete end-to-end example, we consider the simplified setting where the temporal kernel function is given by:

$$K_{\boldsymbol{\theta}_T}(t, t') := k_{\theta_1, \theta_2, \theta_3}(t - t') = \theta_2^2 \frac{2^{1-\theta_1}}{\Gamma(\theta_1)}\left(\sqrt{2\theta_1}\frac{t - t'}{\theta_3}\right)^{\theta_1} B_{\theta_1}\left(\sqrt{2\theta_1}\frac{t - t'}{\theta_3}\right),$$

where $B_{\theta_1}(\cdot)$ is the Bessel function so $K_{\boldsymbol{\theta}_T}(t, t')$ belongs to the well-known Matern family. It is straightforward to show that the spectral density function is given by:

$$s(\omega) \propto \left(\frac{2\theta_1}{\theta_3^2} + \omega^2\right)^{-(\theta_1 + 1/2)}.$$

As a consequence, we see that $s(\omega) \propto \left(\frac{\sqrt{2\theta_1}}{\theta_3} + i\omega\right)^{-(\theta_1+1/2)} \left(\frac{\sqrt{2\theta_1}}{\theta_3} - i\omega\right)^{-(\theta_1+1/2)}$, so we

directly have $G_{\boldsymbol{\theta}_T}(\omega) = \left(\frac{\sqrt{2\theta_1}}{\theta_3} + i\omega\right)^{-(\theta_1+1/2)}$ instead of having to seek for polynomial approximation. Now we can easily expand $G_{\boldsymbol{\theta}_T}(\omega)$ using the binomial formula to find the linear parameters for the continuous-time system. For instance, when $\theta_1 = 3/2$, we have:

$$\frac{\mathrm{d}^2 f(t)}{\mathrm{d}t^2} + 2\frac{\sqrt{2\theta_1}}{\theta_3}\frac{\mathrm{d}f(t)}{\mathrm{d}t} + \frac{2\theta_1}{\theta_3^2}f(t) = \epsilon(t).$$

### A.3.2 PROOF FOR PROPOSITION 1

*Proof.* We first need to show that the random Fourier features for the non-stationary kernel $K_T\big((\mathbf{x},t),(\mathbf{x}',t')\big)$ can be given by (11), i.e.

$$\boldsymbol{\phi}(\mathbf{x},t) = \frac{1}{2\sqrt{m}}\big[\ldots,\cos\big([\mathbf{x},t]^\mathsf{T}\boldsymbol{\omega}_{1,i}\big) + \cos\big([\mathbf{x},t]^\mathsf{T}\boldsymbol{\omega}_{2,i}\big), \sin\big([\mathbf{x},t]^\mathsf{T}\boldsymbol{\omega}_{1,i}\big) + \sin\big([\mathbf{x},t]^\mathsf{T}\boldsymbol{\omega}_{2,i}\big)\ldots\big].$$

To simplify notations, we let $\mathbf{z} := [\mathbf{x},t] \in \mathbb{R}^{d+1}$ and $\mathcal{Z} = \mathcal{X} \times \mathcal{T}$. For non-stationary kernels, their corresponding Fourier transform can be characterized by the following lemma. Assume without loss of generality that $K_T$ is differentiable.

**Lemma A.1** (Yaglom (1987)). *A non-stationary kernel $k(\mathbf{z}_1, \mathbf{z}_2)$ is positive definite in $\mathbb{R}^d$ if and only if after scaling, it has the form:*

$$k(\mathbf{z}_1, \mathbf{z}_2) = \int \exp\big(i(\boldsymbol{\omega}_1^\mathsf{T}\mathbf{z}_1 - \boldsymbol{\omega}_2^\mathsf{T}\mathbf{z}_2)\big)\mu(d\boldsymbol{\omega}_1, d\boldsymbol{\omega}_2), \tag{A.12}$$

*where $\mu(d\boldsymbol{\omega}_1, d\boldsymbol{\omega}_2)$ is some positive-semidefinite probability measure with bounded variation.*

The above lemma can be think of as the extension of the classical Bochner's theorem underlies the random Fourier feature for stationary kernels. Notice that when covariance function for the measure $\mu$ only has non-zero diagonal elements and $\boldsymbol{\omega}_1 = \boldsymbol{\omega}_2$, then we recover the spectral representation stated in the Bochner's theorem. Therefore, we can also approximate (A.12) with the Monte Carlo integral. However, we need to ensure the positive-semidefiniteness of the spectral density for $\mu(d\boldsymbol{\omega}_1, d\boldsymbol{\omega}_2)$, which we denote by $p(\boldsymbol{\omega}_1, \boldsymbol{\omega}_2)$. It has been suggested in Remes et al. (2017) that we consider another density function $q(\boldsymbol{\omega}_1, \boldsymbol{\omega}_2)$ and let $p$ be taken on the product space of $q$ and then symmetrise:

$$p(\boldsymbol{\omega}_1, \boldsymbol{\omega}_2) = \frac{1}{4}\big(q(\boldsymbol{\omega}_1, \boldsymbol{\omega}_2) + q(\boldsymbol{\omega}_2, \boldsymbol{\omega}_1) + q(\boldsymbol{\omega}_1, \boldsymbol{\omega}_1) + q(\boldsymbol{\omega}_2, \boldsymbol{\omega}_2)\big). \tag{A.13}$$

Then (A.12) suggests that

$$k(\mathbf{z}_1, \mathbf{z}_2) = \frac{1}{4}\mathbb{E}_q\Big[\exp\big(i(\boldsymbol{\omega}_1^\mathsf{T}\mathbf{z}_1 - \boldsymbol{\omega}_2^\mathsf{T}\mathbf{z}_2)\big) + \exp\big(i(\boldsymbol{\omega}_2^\mathsf{T}\mathbf{z}_2 - \boldsymbol{\omega}_1^\mathsf{T}\mathbf{z}_1)\big)$$
$$+ \exp\big(i(\boldsymbol{\omega}_1^\mathsf{T}\mathbf{z}_1 - \boldsymbol{\omega}_1^\mathsf{T}\mathbf{z}_1)\big) + \exp\big(i(\boldsymbol{\omega}_2^\mathsf{T}\mathbf{z}_2 - \boldsymbol{\omega}_2^\mathsf{T}\mathbf{z}_2)\big)\Big].$$

Recall that the real part of $\exp\big(i(\boldsymbol{\omega}_1^\mathsf{T}\mathbf{z}_1 - \boldsymbol{\omega}_2^\mathsf{T}\mathbf{z}_2)\big)$ is given by $\cos(\boldsymbol{\omega}_1^\mathsf{T}\mathbf{z}_1 - \boldsymbol{\omega}_2^\mathsf{T}\mathbf{z}_2)$. So with the Trigonometric equalities, it is straightforward to verify that $k(\mathbf{z}_1, \mathbf{z}_2) = \mathbb{E}_q\big[\boldsymbol{\phi}(\mathbf{z})^\mathsf{T}\boldsymbol{\phi}(\mathbf{z})\big]$. Hence, the random Fourier features for non-stationary kernel can be given in the form of (11).

Then we show the uniform convergence result as the number of samples goes to infinity when computing $\mathbb{E}_q\big[\boldsymbol{\phi}(\mathbf{z})^\mathsf{T}\boldsymbol{\phi}(\mathbf{z})\big]$ by the Monte Carlo integral. Let $\tilde{\mathcal{Z}} = \mathcal{Z} \times \mathcal{Z}$, so $\tilde{\mathcal{Z}} = \{(\mathbf{x},t,\mathbf{x}',t,) \,\|\, \mathbf{x},\mathbf{x}' \in \mathcal{X}; t,t' \in \mathcal{T}\}$. Since $\mathrm{diam}(\mathcal{X}) = l$ and $\mathcal{T} = [0,t_{\max}]$, we have $\mathrm{diam}(\tilde{\mathcal{Z}}) = l^2 t_{\max}^2$. Let the approximation error be

$$\Delta(\mathbf{z},\mathbf{z}') = \boldsymbol{\phi}(\mathbf{z})^\mathsf{T}\boldsymbol{\phi}(\mathbf{z}') - K_T(\mathbf{z}.\mathbf{z}'). \tag{A.14}$$

The strategy is to use a $\epsilon$-net covering for the input space $\tilde{\mathcal{Z}}$, which would require $N = \big(2l^2 t_{\max}^2/r\big)^{d+1}$ balls of radius $r$. Let $\mathcal{C} = \{\mathbf{c}_i\}_{i=1}^N$ be the centers for each $\epsilon$-ball. We first show the bound for $|\Delta(\mathbf{c}_i)|$ and the Lipschitz constant $L_\Delta$ of the error function $\Delta$, and then combine them to get the desired result.

Since $\Delta$ is continuous and differentiable w.r.t $\mathbf{z}, \mathbf{z}'$ according to the definition of $\phi$, we have $L_\Delta = \left\|\nabla\Delta(\mathbf{c}^*)\right\|$, where $\mathbf{c}^* = \arg\max_{\mathbf{c}\in\mathcal{C}} \left\|\nabla\Delta(\mathbf{c})\right\|$. Let $\mathbf{c}^* = (\tilde{\mathbf{z}}, \tilde{\mathbf{z}}')$. By checking the regularity conditions for exchanging the integral and differential operation, we verify that $\mathbb{E}\left[\nabla\phi(\mathbf{z})^\intercal\phi(\mathbf{z}')\right] = \nabla\mathbb{E}\left[\phi(\mathbf{z})^\intercal\phi(\mathbf{z}')\right] = \nabla\mathbb{E}\left[K_T(\mathbf{z}, \mathbf{z}')\right]$. We do not present the details here, since it is easy to check the regularity of $\phi(\mathbf{z})^\intercal\phi(\mathbf{z}')$ as it consists of the sine and cosine functions who are continuous, bounded and have continuous bounded derivatives. Hence, we have:

$$
\begin{aligned}
\mathbb{E}\left[L_\Delta^2\right] &= \mathbb{E}_{\tilde{\mathbf{z}},\tilde{\mathbf{z}}'}\left[\left\|\nabla\phi(\tilde{\mathbf{z}})^\intercal\phi(\tilde{\mathbf{z}}') - \nabla K_T(\tilde{\mathbf{z}}, \tilde{\mathbf{z}}')\right\|^2\right] \\
&= \mathbb{E}_{\tilde{\mathbf{z}},\tilde{\mathbf{z}}'}\left[\mathbb{E}\|\nabla\phi(\tilde{\mathbf{z}})^\intercal\phi(\tilde{\mathbf{z}}')\|^2 - 2\|\nabla K_T(\tilde{\mathbf{z}}, \tilde{\mathbf{z}}')\| \cdot \|\nabla\phi(\tilde{\mathbf{z}})^\intercal\phi(\tilde{\mathbf{z}}')\| + \|\nabla K_T(\tilde{\mathbf{z}}, \tilde{\mathbf{z}}')\|^2\right] \\
&\leq \mathbb{E}_{\tilde{\mathbf{z}},\tilde{\mathbf{z}}'}\left[\mathbb{E}\|\nabla\phi(\tilde{\mathbf{z}})^\intercal\phi(\tilde{\mathbf{z}}')\|^2 - \|\nabla K_T(\tilde{\mathbf{z}}, \tilde{\mathbf{z}}')\|^2\right] \text{ (by Jensen's inequality)} \\
&\leq \mathbb{E}\|\nabla\phi(\tilde{\mathbf{z}})^\intercal\phi(\tilde{\mathbf{z}}')\|^2 \\
&= \mathbb{E}\Big\|\nabla\big(\cos(\tilde{\mathbf{z}}^\intercal\boldsymbol{\omega}_1) + \cos(\tilde{\mathbf{z}}^\intercal\boldsymbol{\omega}_2)\big)\big(\cos((\tilde{\mathbf{z}}')^\intercal\boldsymbol{\omega}_1) + \cos((\tilde{\mathbf{z}}')^\intercal\boldsymbol{\omega}_2)\big) \\
&\qquad + \big(\sin(\tilde{\mathbf{z}}^\intercal\boldsymbol{\omega}_1) + \sin(\tilde{\mathbf{z}}^\intercal\boldsymbol{\omega}_2)\big)\big(\sin((\tilde{\mathbf{z}}')^\intercal\boldsymbol{\omega}_1) + \sin((\tilde{\mathbf{z}}')^\intercal\boldsymbol{\omega}_2)\big)\Big\|^2 \\
&= 2\mathbb{E}\Big\|\boldsymbol{\omega}_1\big(\sin(\tilde{\mathbf{z}}^\intercal\boldsymbol{\omega}_1 - (\tilde{\mathbf{z}}')^\intercal\boldsymbol{\omega}_1) + \sin((\tilde{\mathbf{z}}')^\intercal\boldsymbol{\omega}_2 - \tilde{\mathbf{z}}^\intercal\boldsymbol{\omega}_1)\big) \\
&\qquad + \boldsymbol{\omega}_2\big(\sin(\tilde{\mathbf{z}}^\intercal\boldsymbol{\omega}_1 - (\tilde{\mathbf{z}}')^\intercal\boldsymbol{\omega}_2) + \sin((\tilde{\mathbf{z}}')^\intercal\boldsymbol{\omega}_2 - \tilde{\mathbf{z}}^\intercal\boldsymbol{\omega}_2)\big)\Big\|^2 \\
&\leq 8\mathbb{E}\big\|[\boldsymbol{\omega}_1, \boldsymbol{\omega}_2]\big\|^2 = 8\sigma_p^2.
\end{aligned}
\tag{A.15}
$$

Hence, by the Markov's inequality, we have

$$
p\Big(L_\Delta \geq \frac{\epsilon}{2r}\Big) \leq 8\sigma_p^2\Big(\frac{2r}{\epsilon}\Big).
\tag{A.16}
$$

Then we notice that for all $c \in \mathcal{C}$, $\Delta(c)$ is the mean of $m/2$ terms bounded by $[-1, 1]$, and the expectation is 0. So applying a union bound and the Hoeffding's inequality on bounded random variables, we have:

$$
p\Big(\cup_{i=1}^N |\Delta(c_i)| \geq \frac{\epsilon}{2}\Big) \leq 2N\exp\Big(-\frac{m\epsilon^2}{16}\Big).
\tag{A.17}
$$

Combining the above results, we get

$$
\begin{aligned}
p\Big(\sup_{(\mathbf{z},\mathbf{z}')\in\mathcal{C}} \big|\Delta(\mathbf{z}, \mathbf{z}')\big| \leq \epsilon\Big) &\geq 1 - \frac{32\sigma_p^2 r^2}{\epsilon^2} - 2r^{-(d+1)}\Big(\frac{2l^2 t_{\max}}{r}\Big)^{d+1}\exp\Big(-\frac{m\epsilon^2}{16}\Big) \\
&\geq C(d)\Big(\frac{l^2 t_{\max}^2\sigma_p}{\epsilon}\Big)^{2(d+1)/(d+3)}\exp\Big(-\frac{m\epsilon^2}{8(d+3)}\Big),
\end{aligned}
\tag{A.18}
$$

where in the second inequality we optimize over $r$ such that $r^* = \left(\frac{(d+1)k_1}{k_2}\right)^{1/(d+3)}$ with $k_1 = 2(2l^2 t_{\max}^2)^{d+1}\exp(-m\epsilon^2/16)$ and $k_2 = 32\sigma_p^2\epsilon^{-2}$. The constant term is given by $C(d) = 2^{\frac{7d+9}{d+3}}\left(\left(\frac{d+1}{2}\right)^{\frac{-d-1}{d+3}} + \left(\frac{d}{2}\right)^{\frac{2}{d+3}}\right)$. $\qquad\square$

### A.3.3 THE GAUSSIAN PROCESS BEHAVIOR AND NEURAL TANGENT KERNEL AFTER COMPOSING WITH TEMPORAL KERNEL WITH THE RANDOM FEATURE REPRESENTATION

This section is dedicated to show the infinite-width Gaussian process behavior and neural tangent kernel properties, similar to what we discussed in Appendix A.2, when composing neural networks in the feature space with the random feature representation of the temporal kernel.

For brevity, we still consider the standard L-layer FFN of (A.1). Suppose we compose the FFN with the random feature representation $\phi(\mathbf{x}, t)$ at the $k^{th}$ layer. It is easy to see that the neural network

kernel for the first $k - 1$ layer are unchanged, so we compute them in the usual way as in (A.2). For the $k^{th}$ layer, it is straightforward to verify that:

$$\lim_{d_k \to \infty} \mathbb{E}\left[\frac{1}{d_k}\left\langle \mathbf{W}^{(k)} \boldsymbol{f}^{(k-1)}(\boldsymbol{\theta}, \mathbf{x}) \circ \boldsymbol{\phi}(\mathbf{x}, t), \mathbf{W}^{(k)} \boldsymbol{f}^{(k-1)}(\boldsymbol{\theta}, \mathbf{x}') \circ \boldsymbol{\phi}(\mathbf{x}', t')\right\rangle \Big| \boldsymbol{f}^{(k-1)}\right]$$
$$\to \Sigma^{(k)}(\mathbf{x}, \mathbf{x}') \cdot K_T\big((\mathbf{x}, t), (\mathbf{x}', t')\big).$$

The intuition is that the randomness in $\mathbf{W}$ (thus $\boldsymbol{f}(\boldsymbol{\theta}, .)$) and $\boldsymbol{\phi}(., .)$ are independent, i.e. the former is caused by network parameter initializations and the later is induced by the random features. The covariance functions for the subsequent layers can be derived by induction, e.g. for the $(k + 1)^{th}$ layer we have:

$$\Sigma_T^{(k+1)}\big((\mathbf{x}, t), (\mathbf{x}', t')\big) = \mathbb{E}_{\boldsymbol{f} \sim \mathcal{N}\left(0, \boldsymbol{\Sigma}^{(k)} \otimes \mathbf{K}_T\right)}\left[\sigma\big(\boldsymbol{f}(\mathbf{x}, t)\big)\sigma\big(\boldsymbol{f}(\mathbf{x}', t')\big)\right].$$

In summary, composing the FNN, at any given layer, with the temporal kernel using its random feature representation does not change the infinite-width Gaussian process behavior. The statement is true for all the deep learning models who also have the Gaussian process behavior, which includes most of the standard neural architectures including RNN, CNN and attention mechanism (Yang, 2019).

The derivations for the NTK, however, is more involved since the gradient on all the layers are affected. We summarize the result for the L-layer FFN in the following proposition and provide the derivations afterwards.

**Proposition A.1.** *Suppose* $\boldsymbol{f}^{(k)}\big(\boldsymbol{\theta}, (\mathbf{x}, t)\big) = vec(\boldsymbol{f}^{(k)}(\boldsymbol{\theta}, \mathbf{x}) \circ \boldsymbol{\phi}(\mathbf{x}, t))$ *in the standard L-layer FFN.* *Let* $\boldsymbol{\Sigma}_T^{(h)} = \boldsymbol{\Sigma}^{(h)}$ *for* $h = 1, \dots, k$, $\boldsymbol{\Sigma}_T^{(k)} = \boldsymbol{\Sigma}_T^{(k)} \otimes \mathbf{K}_T$ *and* $\boldsymbol{\Sigma}_T^{(h)} = \mathbb{E}_{\boldsymbol{f} \sim \mathcal{N}(0, \boldsymbol{\Sigma}_T^{(h)})}[\sigma(\boldsymbol{f})\sigma(\boldsymbol{f})] + 1$ *for* $h = k + 1, \dots, L$. *If the activation functions* $\sigma$ *have polynomially bounded weak derivatives, as the network widths* $d_1, \dots, d_L \to \infty$, *the neural tangent kernel* $\boldsymbol{\Theta}^{(L)}$ *converges almost surely to* $\boldsymbol{\Theta}_T^{(L)}$ *whose partial application on parameters* $\{\mathbf{W}^{(h)}, \mathbf{b}^{(h)}\}$ *in the* $h^{th}$-*layer is given recursively by:*

$$\boldsymbol{\Theta}_T^{(h)} = \boldsymbol{\Sigma}_T^{(h)}, \quad \boldsymbol{\Theta}_T^{(k)} = \boldsymbol{\Theta}_T^{(k-1)} \otimes \dot{\boldsymbol{\Sigma}}_T^{(k)} + \boldsymbol{\Sigma}_T^{(k)}, \quad k = h + 1, \dots, L. \tag{A.19}$$

*Proof.* The strategies for deriving the NTK and show the convergence has been discussed in Jacot et al. (2018); Yang (2019); Arora et al. (2019a). The key purpose for us presenting the derivations here is to show how the convergence results for the neural-temporal Gaussian (Section 4.2) affects the NTK. To avoid the cumbersome notations induced by the peripheral intercept terms, here we omit the intercept terms $\mathbf{b}$ in the FFN without loss of generality. We let $\mathbf{g}^{(h)} = \frac{1}{\sqrt{d_h}}\sigma\big(\boldsymbol{f}^{(h)}(\mathbf{x}, t)\big)$, so the FFN can be equivalently defined via the recursion: $\boldsymbol{f}^{(h)} = \mathbf{W}^{(h)}\mathbf{g}^{(h-1)}(\mathbf{x}, t)$. For the final output $f\big(\boldsymbol{\theta}, (\mathbf{x}, t)\big) := \mathbf{W}^{(L)}\boldsymbol{f}^{(L)}(\mathbf{x}, t)$, the partial derivative to $\mathbf{W}^{(h)}$ can be given by:

$$\frac{\partial f\big(\boldsymbol{\theta}, (\mathbf{x}, t)\big)}{\partial \mathbf{W}^{(h)}} = \mathbf{z}^{(h)}(\mathbf{x}, t)\big(\mathbf{g}^{(h-1)}(\mathbf{x}, t)\big)^\mathsf{T}, \tag{A.20}$$

with $\mathbf{z}^{(h)}$ defined by;

$$\mathbf{z}^{(h)}(\mathbf{x}, t) = \begin{cases} 1, & h = L, \\ \frac{1}{\sqrt{d_h}}\mathbf{D}^{(h)}(\mathbf{x}, t)\big(\mathbf{W}^{(h+1)}\big)^\mathsf{T}\mathbf{z}^{(h+1)}(\mathbf{x}, t), & h = 1, \dots, L - 1, \end{cases} \tag{A.21}$$

where

$$\mathbf{D}^{(h)}(\mathbf{x}, t) = \begin{cases} \text{diag}\big(\dot{\sigma}\big(\boldsymbol{f}^{(h)}(\mathbf{x}, t)\big)\big), & h = k, \dots, L - 1, \\ \text{diag}\big(\dot{\sigma}\big(\boldsymbol{f}^{(h)}(\mathbf{x})\big)\big), & h = 1, \dots, k - 1. \end{cases}$$

Using the above definitions, we have:

$$\left\langle \frac{\partial f\big(\boldsymbol{\theta}, (\mathbf{x}, t)\big)}{\partial \mathbf{W}^{(h)}}, \frac{\partial f\big(\boldsymbol{\theta}, (\mathbf{x}', t')\big)}{\partial \mathbf{W}^{(h)}} \right\rangle = \left\langle \mathbf{z}^{(h)}(\mathbf{x}, t)\big(\mathbf{g}^{(h-1)}(\mathbf{x}, t)\big)^\mathsf{T}, \mathbf{z}^{(h)}(\mathbf{x}', t')\big(\mathbf{g}^{(h-1)}(\mathbf{x}', t')\big)^\mathsf{T} \right\rangle$$
$$= \left\langle \mathbf{g}^{(h-1)}(\mathbf{x}, t), \mathbf{g}^{(h-1)}(\mathbf{x}', t') \right\rangle \cdot \left\langle \mathbf{z}^{(h)}(\mathbf{x}, t), \mathbf{z}^{(h)}(\mathbf{x}', t') \right\rangle$$

We have established in Section 4.2 that
$$\big\langle \mathbf{g}^{(h-1)}(\mathbf{x}, t), \mathbf{g}^{(h-1)}(\mathbf{x}', t')\big\rangle \to \Sigma_T^{(h-1)}\big((\mathbf{x}, t), (\mathbf{x}', t')\big),$$

where
$$\Sigma_T^{(h)}\big((\mathbf{x}, t), (\mathbf{x}', t')\big) = \begin{cases} \Sigma^{(h)}(\mathbf{x}, \mathbf{x}') & h = 1, \dots, k \\ \Sigma^{(h)}(\mathbf{x}, \mathbf{x}') \cdot K_T\big((\mathbf{x}, t), (\mathbf{x}', t')\big) & h = k \\ \mathbb{E}_{\boldsymbol{f} \sim \mathcal{N}\big(0, \boldsymbol{\Sigma}_T^{(h-1)}\big)}\Big[\sigma\big(\boldsymbol{f}(\mathbf{x}, t)\big)\sigma\big(\boldsymbol{f}(\mathbf{x}', t')\big)\Big] & h = k+1, \dots, L. \end{cases} \tag{A.22}$$

By the definition of $\mathbf{z}^{(h)}$, we get
$$\begin{aligned} &\big\langle \mathbf{z}^{(h)}(\mathbf{x}, t), \mathbf{z}^{(h)}(\mathbf{x}', t')\big\rangle \\ &= \frac{1}{d_h}\big\langle \mathbf{D}^{(h)}(\mathbf{x}, t)\big(\mathbf{W}^{(h+1)}\big)^{\mathsf{T}}\mathbf{z}^{(h+1)}(\mathbf{x}, t), \mathbf{D}^{(h)}(\mathbf{x}', t')\big(\mathbf{W}^{(h+1)}\big)^{\mathsf{T}}\mathbf{z}^{(h+1)}(\mathbf{x}', t')\big\rangle \\ &\approx \frac{1}{d_h}\big\langle \mathbf{D}^{(h)}(\mathbf{x}, t)\big(\mathbf{W}^{(h+1)}\big)^{\mathsf{T}}\mathbf{z}^{(h+1)}(\mathbf{x}, t), \mathbf{D}^{(h)}(\mathbf{x}', t')\big(\tilde{\mathbf{W}}^{(h+1)}\big)^{\mathsf{T}}\mathbf{z}^{(h+1)}(\mathbf{x}', t')\big\rangle \\ &\to \frac{1}{d_h}\mathrm{tr}\Big(\mathbf{D}^{(h)}(\mathbf{x}, t)\mathbf{D}^{(h)}(\mathbf{x}', t')\Big)\big\langle \mathbf{z}^{(h+1)}(\mathbf{x}, t), \mathbf{z}^{(h+1)}(\mathbf{x}', t')\big\rangle \\ &\to \dot{\Sigma}_T^h\big((\mathbf{x}, t), (\mathbf{x}', t')\big\langle \mathbf{z}^{(h+1)}(\mathbf{x}, t), \mathbf{z}^{(h+1)}(\mathbf{x}', t')\big\rangle. \end{aligned} \tag{A.23}$$

The approximation in the third line is made because the $\mathbf{W}^{(h+1)}$ in the right half is replaced by its i.i.d copy under Gaussian initialization. This does not change the limit when $d_h \to \infty$ when the actionvation functions have polynomially bounded weak derivatives Yang (2019) such the ReLU activation. Carrying out (A.23) recursively, we see that
$$\big\langle \mathbf{z}^{(h)}(\mathbf{x}, t), \mathbf{z}^{(h)}(\mathbf{x}', t')\big\rangle \to \prod_{j=h}^{L-1} \dot{\Sigma}^j\big((\mathbf{x}, t), (\mathbf{x}', t')\big).$$

Finally, we have:
$$\begin{aligned} &\Big\langle \frac{\partial f\big(\boldsymbol{\theta}, (\mathbf{x}, t)\big)}{\partial \boldsymbol{\theta}}, \frac{\partial f\big(\boldsymbol{\theta}, (\mathbf{x}', t')\big)}{\partial \boldsymbol{\theta}}\Big\rangle = \sum_{h=1}^{L} \Big\langle \frac{\partial f\big(\boldsymbol{\theta}, (\mathbf{x}, t)\big)}{\partial \mathbf{W}^{(h)}}, \frac{\partial f\big(\boldsymbol{\theta}, (\mathbf{x}', t')\big)}{\partial \mathbf{W}^{(h)}}\Big\rangle \\ &= \sum_{h=1}^{L} \Big(\Sigma_T^{(h)}\big((\mathbf{x}, t), (\mathbf{x}', t')\big) \cdot \prod_{j=h}^{L} \dot{\Sigma}^j\big((\mathbf{x}, t), (\mathbf{x}', t')\big)\Big). \end{aligned} \tag{A.24}$$

Notice that we use a more compact recursive formulation to state the results in Proposition 1. It is easy to verify that after expansion, we reach the desired results. □

Compared with the original NTK before composing with the temporal kernel (given by (A.4)), the results in Proposition A.1 shares a similar recursion structure. As a consequence, the previous results for NTK can be directly adopted to our setting. We list two examples here.

- Following Jacot et al. (2018), given a training dataset $\{\mathbf{x}_i, t_i, y_i(t_i)\}_{i=1}^n$, let $\boldsymbol{f}_T\big(\boldsymbol{\theta}(s)\big) = \big(f(\boldsymbol{\theta}(s), \mathbf{x}_1, t_1), \dots, f(\boldsymbol{\theta}(s), \mathbf{x}_n, t_n)\big)$ be the network outputs at the $s^{th}$ training step and $\mathbf{y}_T = \big(y_1(t_1), \dots, y_n(t_n)\big)$. The analysis on the optimization trajectory under infinitesimal learning rate can be conducted via:
$$\frac{d\boldsymbol{f}_T\big(\boldsymbol{\theta}(s)\big)}{ds} = -\boldsymbol{\Theta}_T(s) \times \big(\boldsymbol{f}_T\big(\boldsymbol{\theta}(s)\big) - \mathbf{y}_T\big),$$
  where $\boldsymbol{\Theta}_T(s)$ converges almost surely to the NTK $\boldsymbol{\Theta}_T^{(L)}$ in Proposition A.1.
- Following Allen-Zhu et al. (2019) and Arora et al. (2019b), the generalization performance of the composed time-aware neural network can be explicitly characterized according to the properties of $\boldsymbol{\Theta}_T^{(L)}$.

### A.3.4 PROOF FOR THEOREM 1

*Proof.* We first present a technical lemma that is crucial for establishing the duality result under the distributional constraint $d_f(S_{\boldsymbol{\theta}_T} \| S) \leq \delta$. Recall that the hidden dimension for the $k^{th}$ layer is $d_k$.

**Lemma A.2** (Ben-Tal et al. (2013)). *Let $f$ be any closed convex function with domain $[0, +\infty)$, and this conjugate is given by $f^*(s) = \sup_{t \geq 0}\{ts - f(t)\}$. Then for any distribution $S$ and any function $g : \mathbb{R}^{d_k + 1} \to \mathbb{R}$, we have*

$$\sup_{S_{\boldsymbol{\theta}_T} : d_f(S_{\boldsymbol{\theta}_T} \| S) \leq \delta} \int g(\boldsymbol{\omega}) dS_{\boldsymbol{\theta}_T}(\boldsymbol{\omega}) = \inf_{\lambda \geq 0, \eta} \left\{ \lambda \int f^*\left(\frac{g(\boldsymbol{\omega}) - \eta}{\lambda}\right) dS(\boldsymbol{\omega}) + \delta\lambda + \eta \right\}. \quad (A.25)$$

We work with a scaled version of the f-divergence under $f(t) = \frac{1}{k}(t^k - 1)$ (because its dual function has a cleaner form), where the constraint set is now equivalent to $\{S_{\boldsymbol{\theta}_T} : d_f(S_{\boldsymbol{\theta}_T} \| S) \leq \delta/k\}$. It is easy to check that $f^*(s) = \frac{1}{k'}[s]_+^{k'} + \frac{1}{k}$ with $\frac{1}{k'} + \frac{1}{k} = 1$.

Similar to the proof for Proposition 1, we let $\mathbf{z} := [\mathbf{x}, t] \in \mathbb{R}^{d+1}$ and $\mathcal{Z} = \mathcal{X} \times \mathcal{T}$ to simplify the notations. To explicitly annotate the dependency of the random Fourier features on $\boldsymbol{\Omega}$, which is the random variable corresponding to $\boldsymbol{\omega}$, we define $\tilde{\phi}(\mathbf{z}, \boldsymbol{\Omega})$ such that $\tilde{\phi}(\mathbf{z}, \boldsymbol{\Omega}) = \big[\cos(\mathbf{z}^\intercal \boldsymbol{\Omega}_1) + \cos(\mathbf{z}^\intercal \boldsymbol{\Omega}_2), \sin(\mathbf{z}^\intercal \boldsymbol{\Omega}_1) + \sin(\mathbf{z}^\intercal \boldsymbol{\Omega}_2)\big]$, where $\boldsymbol{\Omega} = [\boldsymbol{\Omega}_1, \boldsymbol{\Omega}_2]$. Then the approximation error, when replacing the sampled Fourier features $\phi$ by the original random variable $\tilde{\phi}(\mathbf{z}, \boldsymbol{\Omega})$, is given by:

$$\begin{aligned}
\Delta_n(\boldsymbol{\Omega}) &:= \frac{1}{n(n-1)} \sum_{i \neq j} \Sigma^{(k)}(\mathbf{x}_i, \mathbf{x}_j)\tilde{\phi}(\mathbf{z}_i, \boldsymbol{\Omega})^\intercal \tilde{\phi}(\mathbf{z}_j, \boldsymbol{\Omega}) \\
&\qquad - \mathbb{E}\big[\Sigma^{(k)}(\mathbf{X}_i, \mathbf{X}_j) K_{T, S_{\boldsymbol{\theta}_T}}\big((\mathbf{X}_i, T_i), (\mathbf{X}_j, T_j)\big)\big] \\
&= \frac{1}{n(n-1)} \sum_{i \neq j} \Sigma^{(k)}(\mathbf{x}_i, \mathbf{x}_j)\tilde{\phi}(\mathbf{z}_i, \boldsymbol{\Omega})^\intercal \tilde{\phi}(\mathbf{z}_j, \boldsymbol{\Omega}) - \mathbb{E}\big[\Sigma^{(k)}(\mathbf{X}, \mathbf{X}')\tilde{\phi}(\mathbf{Z}, \boldsymbol{\Omega})^\intercal \tilde{\phi}(\mathbf{Z}', \boldsymbol{\Omega})\big].
\end{aligned}$$

$$(A.26)$$

We first show that sub-Gaussianity of $\Delta_n(\boldsymbol{\Omega})$. Let $\{\mathbf{x}_i'\}_{i=1}^n$ be an i.i.d copy of the observations except for one element $j$ such that $\mathbf{x}_j \neq \mathbf{x}_j'$. Without loss of generality, we assume the last element is different, i.e. $\mathbf{x}_n \neq \mathbf{x}_n'$. Let $\Delta_n'(\boldsymbol{\Omega})$ be computed by replacing $\mathbf{x}$ and $\mathbf{z}$ with the above $\mathbf{x}'$ and its corresponding $\mathbf{z}'$. Note that

$$\begin{aligned}
&|\Delta_n(\boldsymbol{\Omega}) - \Delta_n'(\boldsymbol{\Omega})| \\
&= \frac{1}{n(n-1)}\Big|\sum_{i \neq j} \Sigma^{(k)}(\mathbf{x}_i, \mathbf{x}_j)\tilde{\phi}(\mathbf{z}_i, \boldsymbol{\Omega})^\intercal \tilde{\phi}(\mathbf{z}_j, \boldsymbol{\Omega}) - \Sigma^{(k)}(\mathbf{x}_i', \mathbf{x}_j')\tilde{\phi}(\mathbf{z}_i', \boldsymbol{\Omega})^\intercal \tilde{\phi}(\mathbf{z}_j', \boldsymbol{\Omega})\Big| \\
&\leq \frac{1}{n(n-1)}\Big(\sum_{i < n} \big|\Sigma^{(k)}(\mathbf{x}_i, \mathbf{x}_n)\tilde{\phi}(\mathbf{z}_i, \boldsymbol{\Omega})^\intercal \tilde{\phi}(\mathbf{z}_n, \boldsymbol{\Omega}) - \Sigma^{(k)}(\mathbf{x}_i, \mathbf{x}_n')\tilde{\phi}(\mathbf{z}_i, \boldsymbol{\Omega})^\intercal \tilde{\phi}(\mathbf{z}_n', \boldsymbol{\Omega})\big| \\
&\qquad\qquad + \sum_{j < n} \big|\Sigma^{(k)}(\mathbf{x}_n, \mathbf{x}_j)\tilde{\phi}(\mathbf{z}_n, \boldsymbol{\Omega})^\intercal \tilde{\phi}(\mathbf{z}_j, \boldsymbol{\Omega}) - \Sigma^{(k)}(\mathbf{x}_n', \mathbf{x}_j)\tilde{\phi}(\mathbf{z}_n', \boldsymbol{\Omega})^\intercal \tilde{\phi}(\mathbf{z}_j, \boldsymbol{\Omega})\big|\Big) \\
&\leq \frac{4\max\{1, M\}}{n},
\end{aligned}$$

$$(A.27)$$

where the last inequality comes from the fact that the random Fourier features $\tilde{\phi}$ are bounded by 1 and the infinity norm of $\boldsymbol{\Sigma}^{(k)}$ is bounded by $M$. The above bounded difference property suggests that $\Delta_n(\boldsymbol{\Omega})$ is a $\frac{4\max\{1, M\}}{n}$-sub-Gaussian random variable.

To bound $\Delta_n(\mathbf{\Omega})$, we use:

$$
\sup_{S_{\boldsymbol{\theta}_T}:d_f(S_{\boldsymbol{\theta}_T}||S)} \left| \int \Delta_n(\mathbf{\Omega})dS_{\boldsymbol{\theta}_T} \right| \leq \sup_{S_{\boldsymbol{\theta}_T}:d_f(S_{\boldsymbol{\theta}_T}||S)} \int |\Delta_n(\mathbf{\Omega})|dS_{\boldsymbol{\theta}_T}
$$

$$
\leq \inf_{\lambda \geq 0} \left\{ \frac{\lambda^{1-k'}}{k'}\mathbb{E}_S\big[|\Delta_n(\mathbf{\Omega})|^{k'}\big] + \frac{\lambda(\delta+1)}{k} \right\} \text{ (using Lemma 2)}
$$

$$
= (\delta+1)^{1/k}\mathbb{E}_S\big[|\Delta_n(\mathbf{\Omega})|^{k'}\big]^{1/k'} \text{ (solving for } \lambda^* \text{ from above)}
$$

$$
= \sqrt{\delta+1}\mathbb{E}_S\big[|\Delta_n(\mathbf{\Omega})|^2\big]^{1/2} \quad \text{(let } k = k' = 1/2).
$$

$$\text{(A.28)}$$

Therefore, to bound $\sup_{S_{\boldsymbol{\theta}_T}:d_f(S_{\boldsymbol{\theta}_T}||S)} \left| \int \Delta_n(\mathbf{\Omega})dS_{\boldsymbol{\theta}_T} \right|$ we simply need to bound $\big[|\Delta_n(\mathbf{\Omega})|^2\big]$. Using the classical results for sub-Gaussian random variables (Boucheron et al., 2013), for $\lambda \leq n/8$, we have

$$
\mathbb{E}\big[ \exp\left(\lambda \Delta_n(\mathbf{\Omega})\right)^2\big] \leq \exp\left( -\frac{1}{2}\log(1 - 8\max\{1, M\}\lambda/n)\right).
$$

Then we take the integral over $\boldsymbol{\omega}$

$$
p\Big( \int \Delta_n(\boldsymbol{\omega})^2 dS(\boldsymbol{\omega}) \geq \frac{\epsilon^2}{\delta+1} \Big)
$$

$$
\leq \mathbb{E}\Big[ \exp\Big( \lambda \int \Delta_n(\boldsymbol{\omega})^2 dS(\boldsymbol{\omega})\Big)\Big] \exp\Big( -\frac{\lambda\epsilon^2}{\delta+1} \Big) \quad \text{(Chernoff bound)} \qquad \text{(A.29)}
$$

$$
\leq \exp\Big( -\frac{1}{2}\log\Big(1 - \frac{8\max\{1, M\}\lambda}{n}\Big) - \frac{\lambda\epsilon^2}{\delta+1} \Big) \quad \text{(apply Jensen's inequality)}.
$$

Finally, let the true approximation error be $\hat{\Delta}_n(\boldsymbol{\omega}) = \hat{\Sigma}^{(k)}(S_{\boldsymbol{\theta}_T}) - \Sigma^{(k)}(S_{\boldsymbol{\theta}_T})$. Notice that

$$
\big|\hat{\Delta}_n(\boldsymbol{\omega})\big| \leq \big|\Delta_n(\mathbf{\Omega})\big| + \frac{1}{n(n-1)}\sum_{i \neq j}\Sigma^{(k)}(\mathbf{x}_i, \mathbf{x}_j)\big|\tilde{\phi}(\mathbf{z}_i, \mathbf{\Omega})^{\mathsf{T}}\tilde{\phi}(\mathbf{z}_j, \mathbf{\Omega}) - \phi(\mathbf{z}_i)^{\mathsf{T}}\phi(\mathbf{z}_j)\big|.
$$

From (A.28) and (A.29), we are able to bound $\sup_{S_{\boldsymbol{\theta}_T}:d_f(S_{\boldsymbol{\theta}_T}||S)} \Delta_n(\mathbf{\Omega})$. For the second term, recall from Proposition 1 that we have shown the stochastic uniform convergence bound for $\big|\tilde{\phi}(\mathbf{z}_i, \mathbf{\Omega})^{\mathsf{T}}\tilde{\phi}(\mathbf{z}_j, \mathbf{\Omega}) - \phi(\mathbf{z}_i)^{\mathsf{T}}\phi(\mathbf{z}_j)\big|$ under any distributions $S_{\boldsymbol{\theta}_T}$. The desired bound for $p\Big( \sup_{S_{\boldsymbol{\theta}_T}:d_f(S_{\boldsymbol{\theta}_T}||S)} \big|\hat{\Delta}_n(\boldsymbol{\omega})\big| \geq \epsilon \Big)$ is obtained after combining all the above results. $\qquad \square$

### A.3.5 REPARAMETRIZATION WITH INVERTIBLE NEURAL NETWORK

In this part, we discuss the idea of constructing and sampling from an arbitrarily complex distribution from a known auxiliary distribution by a sequence of invertible transformations. Given an auxiliary random variable $\mathbf{z}$ following some know distribution $q(z)$, suppose another random variable $\mathbf{x}$ is constructed via a one-to-one mapping from $\mathbf{z}$: $\mathbf{x} = f(\mathbf{z})$, then the density function of $\mathbf{x}$ is given by:

$$
p(x) = q(z)\Big|\frac{dz}{dx}\Big| = q\big(f^{-1}(x)\big)\Big|\frac{df^{-1}}{dx}\Big|. \qquad \text{(A.30)}
$$

We can parameterize the one-to-one function $f(.)$ with free parameters $\theta$ and optimize them over the observed evidence such as by maximizing the log-likelihood. By stacking a sequence of $Q$ one-to-one mappings, i.e. $\mathbf{x} = f_Q \circ f_{Q-1} \circ \ldots f_1(\mathbf{z})$, we can construct complicated density functions. It is easy to show by chaining that $p(x)$ is given by:

$$
\log p(\mathbf{x}) = \log q(\mathbf{z}) - \sum_{i=1}^{Q}\Big|\frac{df^{-1}}{dz_i}\Big|. \qquad \text{(A.31)}
$$

Samples from the auxiliary distribution can be transformed to the unknown target distribution in the same manner, and the transformed samples are essentially parameterized by the transformation mappings.

Unfortunately, most standard neural architectures are non-invertible, so we settle on a specific family of neural networks - the *invertible neural network (INN)* Ardizzone et al. (2018). A major component of INN is the *affine coupling layer*. With $\mathbf{z}$ sampled from the auxiliary distribution, we first divide $\mathbf{z}$ into two halves $[\mathbf{z}_1, \mathbf{z}_2]$ and then let:

$$
\begin{aligned}
\mathbf{v}_1 &= \mathbf{z}_1 \odot \exp\big(\mathbf{s}_1(\boldsymbol{\gamma}, \mathbf{z}_2)\big) + \mathbf{t}_1(\boldsymbol{\gamma}, \mathbf{z}_2) \\
\mathbf{v}_2 &= \mathbf{z}_2 \odot \exp\big(\mathbf{s}_2(\boldsymbol{\gamma}, \mathbf{z}_1)\big) + \mathbf{t}_2(\boldsymbol{\gamma}, \mathbf{z}_1),
\end{aligned}
\tag{A.32}
$$

where $\mathbf{s}_1(\boldsymbol{\gamma}, \cdot)), \mathbf{s}_1(\boldsymbol{\gamma}, \cdot)), \mathbf{t}_1(\boldsymbol{\gamma}, \cdot)), \mathbf{t}_1(\boldsymbol{\gamma}, \cdot))$ can be any function parameterized by different parts of $\boldsymbol{\gamma}$. Here, $\odot$ denotes the element-wise product. Then the outcome is simply given by:

$$
\mathbf{g}(\boldsymbol{\gamma}, \mathbf{z}) = [\mathbf{v}_1, \mathbf{v}_2].
$$

To see that $\mathbf{g}(\boldsymbol{\gamma}, \cdot)$ is invertible so the inverse transform mappings are tractable, it is straightforward to show that $\mathbf{g}^{-1}(\boldsymbol{\gamma}, [\mathbf{v}_1, \mathbf{v}_2])$ is given by:

$$
\begin{aligned}
\mathbf{z}_2 &= \big(\mathbf{v}_2 - \mathbf{t}_1(\boldsymbol{\gamma}, \mathbf{v}_1)\big) \odot \exp\big(-\mathbf{s}_1(\boldsymbol{\gamma}, \mathbf{v}_1)\big) \\
\mathbf{z}_1 &= \big(\mathbf{v}_1 - \mathbf{t}_2(\boldsymbol{\gamma}, \mathbf{v}_2)\big) \odot \exp\big(-\mathbf{s}_2(\boldsymbol{\gamma}, \mathbf{v}_2)\big).
\end{aligned}
\tag{A.33}
$$

By stacking multiple affine coupling layers, scalar multiplication and summation actions (which are all invertible), we are able to construct an INN with enough complexity to characterize any non-degenerating distribution.

## B   SUPPLEMENTARY MATERIAL FOR SECTION 5

We provide the detailed dataset description, experiment setup, model configuration, parameter tuning, training procedure, validation, testing, sensitivity analysis and model analysis. The reported results are averaged over five iterations.

### B.1   DATASETS

- **Jena weather dataset**[1]. The dataset contains 14 different features such as air temperature, atmospheric pressure, humidity, and other metrics that reflect certain aspect of the weather. The data were collected between between 2009 and 2016 for every 10 minutes, so there are 6 observations in each hour.

  A standard task on this dataset is to use 5 days of observations to predict the temperature 12 hours in the future, which we refer to as the **Case 1**. We use a sliding window to obtain the training, validation and testing samples and make sure they have no overlaps (right panel of Figure A.2).

- **Wikipedia traffic.**[2] The Wiki web page traffic records the daily number of visits for 550 Wikipedia pages from 2015-07-01 to 2016-12-31. The features are decoded from the webpage url, where we are able to obtain the *project name*, e.g. zh.wikipedia.org, the *access*, e.g. all-access, and the *agent*, e.g. spider. We use one-hot encoding to represent the features, and end up with a 14-dimension feature vector for each webpage.

  The feature vectors do not change with time. We use the feature vectors and traffic data from the past 200 days to predict the traffic of the next 14 days, which is also a standard task for this dataset. The missing data are treated as zero.

- **Alibaba online shopping data.**[3]. The dataset contains 4,136 online shopping sequences with a total of 1,552 items. Each shopping sequence has a varying number of time-stamped user-item interactions, from 11 to 157. We consider the standard next-item recommendation task, where we make a recommendation based on the past interactions. No user or item features are available.

- **Walmart.com e-commerce data.**[4] The session-based online shopping data contains $\sim$12,000 shopping sessions made by 1,000 frequent users, with a total of 2,034 items.

---

[1]https://www.bgc-jena.mpg.de/wetter/

[2]https://www.kaggle.com/c/web-traffic-time-series-forecasting/data

[3]https://github.com/musically-ut/tf_rmtpp/tree/master/data/real/ali

[4]https://github.com/StatsDLMathsRecomSys/Inductive-representation-learning-on-temporal-graphs

The lengths of the sessions vary from 14 to 87. In order to be consistent with the Alibaba online shopping data, we do not use the provided item and user features. We also consider the next-item recommendation task.

**Preprocessing, train-validation-test split and metric** To ensure fair comparisons across the various models originated from different setting, we minimize the data-specific preprocessing steps, especially for the time series dataset.

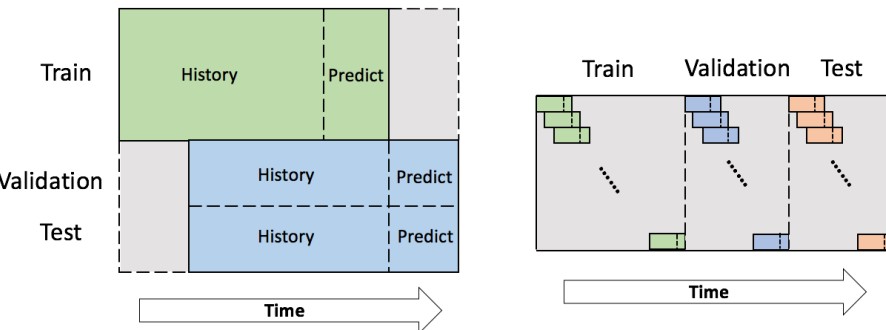

Figure A.2: **Left**: the walk-forward split. Notice that we do not to specify the train-validation proportions when using the walk-forward split. **Right**: side-by-side split with moving window.

- **Jena weather dataset**. We do the train-validation-test split by 60%-20%-20% on the time axis (right panel of Figure A.2). We first standardize the features on the training data, and then use the mean and standard deviation computed on the training data to standardize the validation and testing data, so there is no information leak.

  For **Case 1**, we use the observations made at each hour (one every six observations) in the most recent 120 hours (5 days) to predict the temperature 12 hours in the future.

  For **Case 2**, we randomly sample 120 observations from the most recent 120 hours (with a total of 720 observations), to predict the temperature 12 hours in the future.

  For **Case 3**, we randomly sample 120 observations from the most recent 120 hours (with a total of 720 observations), to predict the temperature randomly sampled from 4 to 12 hours (with a total of 48 observations) in the future.

- **Wikipedia traffic.** We use the walk-forward split (illustrated in the left panel of Figure A.2) to test the performance of the proposed approaches under different train-validation-test split schema. The walk-forward split is helpful when the length of training time series is relatively long compared with the full time series. For instance, on the Wiki traffic data, the full sequence length is 500, and the training sequence length is 200, so it is impossible to conduct the side-by-side split. The features are all one-hot encoding, so we do not carry out any preprocessing. The web traffics are standardized in the same fashion as the Jena weather data.

  For **Case 1**, we use the most recent 200 observations to predict the full web traffics for 14 days in the future.

  For **Case 2**, we randomly sample 200 observations from the past 500 days (with a total of 500 observations), to predict the full web traffics for 14 days in the future.

  For **Case 3**, we randomly sample 200 observations from the past 500 days (with a total of 500 observations), to predict 6 web traffics sampled from 0 to 14 days in the future.

- **Alibaba data.** We conduct the standard preprocessing steps used in the recommender system literature. We first filter out items that have less than 5 total occurrences, and then filter out shopping sequences that has less then 10 interactions. Using the standard train-validation-test split in sequential recommendation, for each shopping sequence, we use all but the last two records as training data, the second-to-last record as validation data, and the last record as testing data.

  All the training/validation/testing samples obtained from the real shopping sequences are treated as the positive record. For each positive record $\{x(t-k), \ldots, x(t), x(t+1)\}$, we

randomly sample 100 items $\{x_i'(t+1)\}_{i=1}^{100}$, and treat each $(x(t-k), \ldots, x(t), x_i'(t+1)')$ as a negative sample, which is again a standard implementation in the recommender systems where no negative labels are available.

- **Walmart.com data.** We use the same preprocessing steps and train-validation-test split as the Alibaba data.

As for the **metrics**, we use the standard **Mean absolute error (MAE)** for the time-series prediction tasks. For the item recommendation tasks, we use the information retrieval metrics **accuracy** and **discounted cumulative gain (DCG)**. Recall that for each shopping sequence, there is one positive sample and 100 negative samples. We rank the candidates $\{x(t+1), x_1'(t+1), \ldots, x_{100}'(t+1)\}$ according to the model's output score on each item. The **accuracy** checks whether the candidate with the highest score is the positive $x(t+1)$, and the **DCG** is given by $1/\log\big(\mathrm{rank}\big(x(t+1)\big)\big)$ where $\mathrm{rank}\big(x(t+1)\big)$ is the position at which the positive $x(t+1)$ is ranked.

## B.2 MODEL CONFIGURATION AND IMPLEMENTATION DETAIL

We observe from the empirical results that using kernel addition ($\Sigma_T^{(h)}(\mathbf{x}, t; \mathbf{x}', t') = \Sigma^{(h)}(\mathbf{x}, \mathbf{x}') + K_T(\mathbf{x}, t; \mathbf{x}', t')$) and kernel multiplication give similar results in our experiments, but kernel addition is much faster in both training and inference. Therefore, we choose to use kernel addition as a trick to expedite the computation. Note that kernel addition corresponds to simply adding up the hidden representations from neural network, and the random features from the temporal kernel.

We first show the configuration and implementation for the models we use in time-series prediction. All the models take the same inputs for each experiment with the hyperparamters tuned on the validation data. Note that the VAR model do not have randomness in model initialization and training, so their outputs do not have standard deviations.

For the **NN+time**, **NN+trigo** and **T-NN** models, the temporal structures (time feature) are added to the same part of the neural architectures (illustrated in Figure A.1), and are made to have the same dimension 32 (except for **NN+time**). We will conduct sensitivity analysis later on the dimension.

For the proposed **T-NN** models, we treat the number of INN (affine coupling) blocks used in building the spectral distribution as tuning parameters. All the remaining model configurations are the same across all model variants. We do not experiment on using regularizations, dropouts or residual connections. In terms of reparameterization, we draw samples from the auxiliary distribution **once** at the beginning, so we do not have to resample during the training process.

- **VAR**. In the vector autoregression model, each variable is modeled as a linear combination of past values of itself and the past values of other variables in the system. The order, i.e. the number of past values used, is treated as the hyperparameter, which we select according according to the AIC criteria on the validation data.

  For experiments on the Jena weather dataset, we choose the order from $\{20, 40, 60, 80, 100, 120\}$, since the maximum length of the history used for prediction is 120 for all three cases. Similarly, for experiments on the Wiki traffic dataset, we choose the order from $\{40, 60, 80, \ldots, 200\}$.

- **RNN models**. We use the one-layer RNN model the standard RNN cells. The hidden dimension for the RNN models is selected to be 32 after tuning on the orignal model. To make time-series prediction, the output of the final state is then passed to a two-layer fully-connected multi-layer perceptron (MLP) with ReLU as the activation function. We adjust the hidden dimensions of the MLP for each model variant such that they have approximately the same number of parameters.

- **CausalCNN models**. We adopt the CausalCNN (Wavenet) architecture from Oord et al. (2016). The original CausalCNN treats the number of filters, filter width, dilation rates and number of convolutional layers are hyperparameters. Here, to avoid the extensive parameter tuning for each model variant, we tune the above parameter of the plain CausalCNN model and adpot them to the other model variants. Specifically, we find that using 16 filters where each filter has a width of 2, with 5 convolutional layers and the dilation rates given by $\{2^l\}_{l=1}^5$, gives the best result for all three cases. Similar to the RNN models, we then pass the output to a two-layer MLP to obtain the prediction.

- **Attention models**. We use a single attention block in the self-attention architecture Vaswani et al. (2017). Unlike the ordinary attention in sequence-to-sequence learning, here we need to employ an extra causal mask (shown in Figure 1) to make sure that the model is not using the future information to predict the past. Other than that, we adopt the same key-query-value attention setting, with their dimension as the tuning parameter. We find that dimension=16 gives the best result in all cases for the original model and we adopt this setting to the rest model variants. Also, we pass the output to a two-layer MLP to obtain the prediction.

We now discuss the configurations and implementations for the recommendation models. We find out that the two-tower architecture illustrated in Figure A.3 is widely used for sequential recommendation (Hidasi et al., 2015; Li et al., 2017a). Each item is first passed through an embedding layer, and the history is processed by some **sequence processing model**, while the target is transformed by some *feed-forward neural network (FFN)* such as the MLP. The outputs from the two towers are then combined together, often by concatenating, and then pass to the top-layer FFN to obtain a prediction score. Hence, we adopt this complex neural architecture to examine our approaches, with the **sequence processing model** in Figure A.3 replaced by their time-aware counterparts given in Figure A.1.

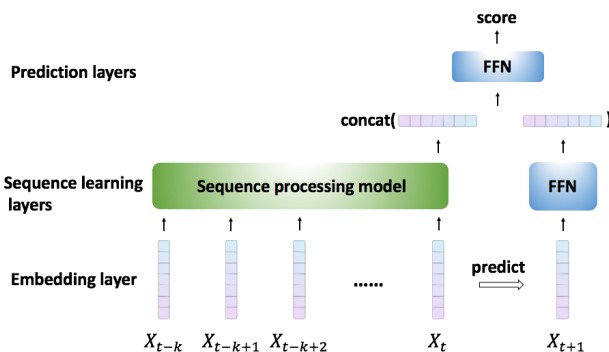

Figure A.3: A standard two-tower neural architecture for sequential recommendation. To incorporate the continuous-time information, we adopt the model architectures in A.1 to replace the **sequence processing model** here.

To be consistent across the Alibaba and Walmart.com dataset, we use the same model configurations. Specifically, the dimension of the embedding layer is chosen to be 100, the FNN on the sequence learning layer is a single-layer MLP with ReLU as activation, and the prediction layer FNN a two-layer MLP with ReLU as activation. We keep the modules to be as simple as possible to avoid overcomplications in model selection. For the **T-NN** models, we treat the dimension of the random Fourier features $\phi$, and the number of INN (affine coupling) blocks used in building the spectral distribution as tuning parameters. We also do not experiment on using regularizations, dropouts or residual connections.

- **GRU-Rec**. We use a single layer RNN with GRU cells as the sequence processing model. We treat the hidden dimension of GRU as a tuning parameter, and according to the validation performance evaluated by the *accuracy*, dimension=32 gives the best outcome for the plain GRUforRec, and we adopt it for all the RNN variants.

- **CNN-Rec**. Similar to the experiments for time-series prediction, we treat the number of convolutional layers, dilation rates, number of filters and filter width as tuning parameters. We select the best settings for the plain CNNforRec and adopt them to all the model variants.

- **ATTN-Rec**. We also use the single-head single-block self-attention model as the sequence processing model. We treat the dimension of the key, query, value matrices (which are of the same dimension) as the turning parameter. It turns out that dimension=20 gives the best validation performance for ATTNforRec, which we adopt to all the model variants.

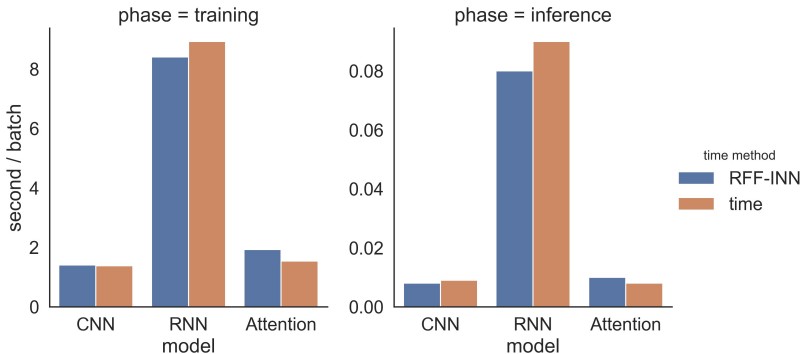

Figure A.4: The **training and inference speed** comparisons for standard neural architectures (RNN, CausalCNN, self-attention) equipped with the proposed temporal kernel approach, and concatenating the time to feature vector.

Unlike the time-series data where the all the samples have a equal sequence length, the shopping sequences have various lengths. Therefore, we set the maximum length to be 100 and use the masking layer to take account of the missing entries.

Finally, we mention the implementation to handle the different scales and formats of the timestamps. The scale of the timespan between events can be very different across datasets, and the absolute value of the timestamps are often less informative, as they could be the linux epoch time or the calendar date. Hence, for each sequence given by $\{(x_1, t_1), \ldots, (x_k, t_k), (x_{k+1}, t_{k+1})\}$ where the target is to predict $(x_{k+1}, t_{k+1})$, we convert it to the representation under timespans: $\{((x_1, t_{k+1} - t_1)), \ldots, (x_k, t_{k+1} - t_k), (x_{k+1}, 0)\}$. We then transform the scale of the timespans to match with the problem setting, for instance, in the online shopping data the timespan is measured by the minute, and in the weather prediction it is measured by the hour.

### B.3 COMPUTATION

The VAR is implemented using the Python module *statsmodels*[5]. The deep learning models are implemented using *Tensorflow 2.0* on a single Nvidia V100 GPU. We use the Adam optimizer and adopt the early-stopping training schema where the trainning is stopped if the validation metric stops improving for 5 epochs. The loss function is **mean absolute error** for the time series prediction, and **binary cross-entropy loss with the softmax function** for the recommendation tasks. The final metrics are computed on the hold-out test data using model checkpoints saved during training that has the best validation performance.

We briefly discuss the computation efficiency of our approach. From Figure A.4 we see that the extra computation time for the proposed temporal kernel approach is almost negligible compared with concatenating time to the feature vector. The advantage is partly due to the fact that we draw samples from the auxiliary distribution only once during training. Note that this does not interfere with our theoretical results, and greatly speeds up the computation.

### B.4 VISUAL RESULTS

We visualize the predictions of our approaches on the Jena weather data in Figure A.5. We see that **T-RNN**, **T-CNN** and **T-ATTN** all capture the temporal signal well. In general, **T-RNN** gives slightly better predictions on the Jena weather dataset.

---

[5]https://www.statsmodels.org/dev/vector_ar.html

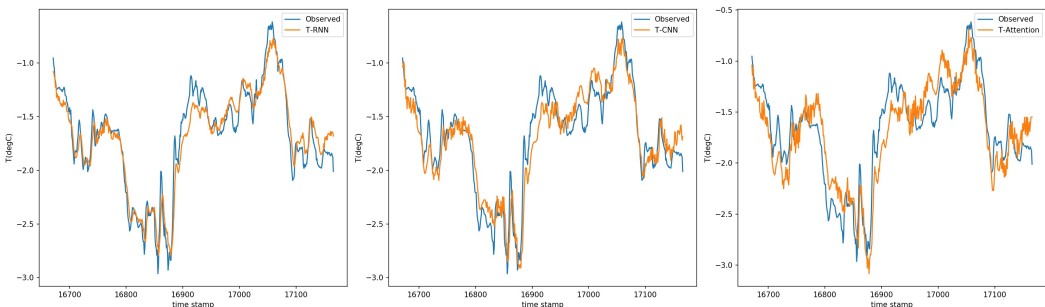

Figure A.5: The predictions of the **T-RNN**, **T-CNN** and **T-ATTN** approaches. We use the models trained for **Case 1** that predict the temperature 12 hours in the future. The plot is made by using sliding windows. The timestamp reflects the hours.

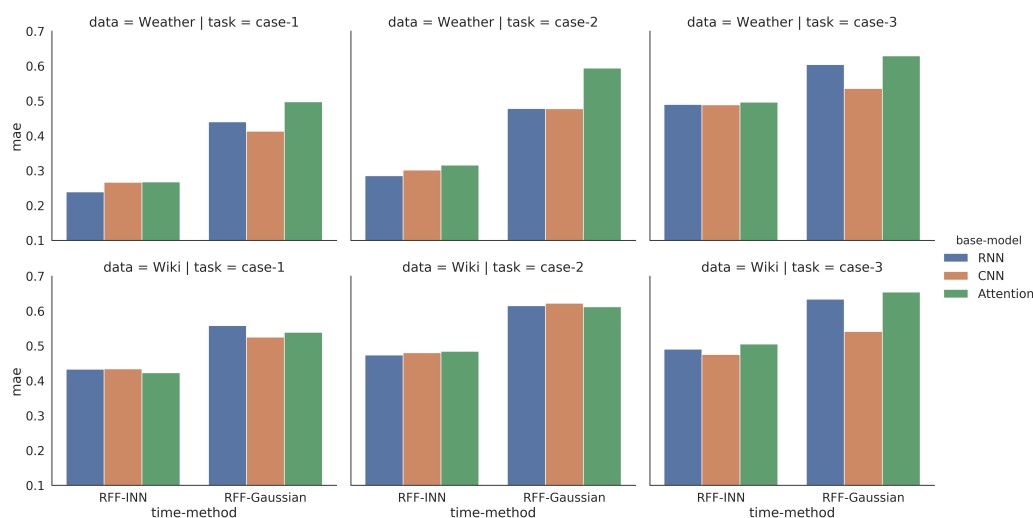

Figure A.6: The ablation study on parameterizing the spectral density with Gaussian distribution, or with INN.

## B.5 EXTRA ABLATION STUDY

We show the effectiveness of using INN to characterize the temporal kernel, compared with using a known distribution family. We focus on the Gaussian distribution for illustration, where both the mean and (diagonal of) the covariance are treated as free parameters. In short, we now parameterize the spectral distribution as Gaussian instead of using INN. We denote this approach by **NN-RF**, to differentiate with the temporal kernel approach **T-NN**.

First, we compare the results between **T-NN** and **NN-RF** on the time-series prediction tasks (shown in Figure A.6). We see that **T-NN** uniformly outperforms **NN-RF**. The results for the recommendation task is similar, as we show in Table A.3, where **T-NN** still achieves the best performance. The numerical results are consistent with Theorem 1 where a more complex distribution family can lead to better outcome. It also justifies our usage of the INN to parameterize the spectral distribution.

## B.6 SENSITIVITY ANALYSIS

We conduct sensitivity analysis to reveal the stability of the proposed approaches with respect to the model selections and our proposed learning schema. Specifically, we focus on the dimension of the random features $\phi$ and the number of INN (affine coupling) blocks when parametrizing the spectral distribution. The results for the time-series prediction tasks are given in Figure A.7, A.8

Table A.3: The ablation study on parameterizing the spectral density with Gaussian distribution, or with INN, when composing the recommendation models with temporal kernel. The reported are the accuracy and *discounted cumulative gain (DCG)* for the domain-specific models on temporal recommendation tasks.

| Metric | Alibaba | | Walmart | |
|---|---|---|---|---|
| | Accuracy | DCG | Accuracy | DCG |
| GRU-Rec-RF | 78.05/.22 | 47.30/.13 | 19.96/.15 | 4.82/.21 |
| **T-GRU-Rec** | *79.47/.35* | *49.82/.40* | *23.41/.11* | *8.44/.21* |
| CNN-Rec-RF | 75.23/.35 | 44.60/.26 | 16.33/.20 | 2.74/.19 |
| **T-CNN-Rec** | *76.45/.16* | *46.55/.38* | *18.59/.33* | *4.56/.31* |
| ATTN-Rec-RF | 52.30/.47 | 31.51/.55 | 22.73/.41 | 7.95/.23 |
| **T-ATTN-Rec** | *53.49/.31* | *33.58/.30* | ***25.51/.17*** | ***9.22/.15*** |

and A.11. The sensitivity analysis results for the recommendation task on Alibaba and Walmart.com datasets are provided in Figure A.9 and A.10. From Table A.3 and A.2, we see that the **ATTN-Rec** models do not perform well on the Aliababa dataset, and the **CNN-Rec** models do not perform well on the Walmart.com dataset. Therefore, we do not provide sensitivty analysis for those models on the corresponding dataset.

In general, the pattern is clear with respect to the dimension of $\phi$, that the larger the dimension (within the range we consider), the better the performance. The result is also reasonable, since a larger dimension for $\phi$ may express more temporal information and better characterize the temporal kernel On the other hand, the pattern for the number of INN (affine coupling layers) blocks is less uniform, where in some cases too few INN blocks suffer from under-fitting, while in some other cases too may INN blocks lead to over-fitting. Therefore, we would recommend using the number of INN blocks as a hyperparameter, and keep the dimension of $\phi$ reasonably large.

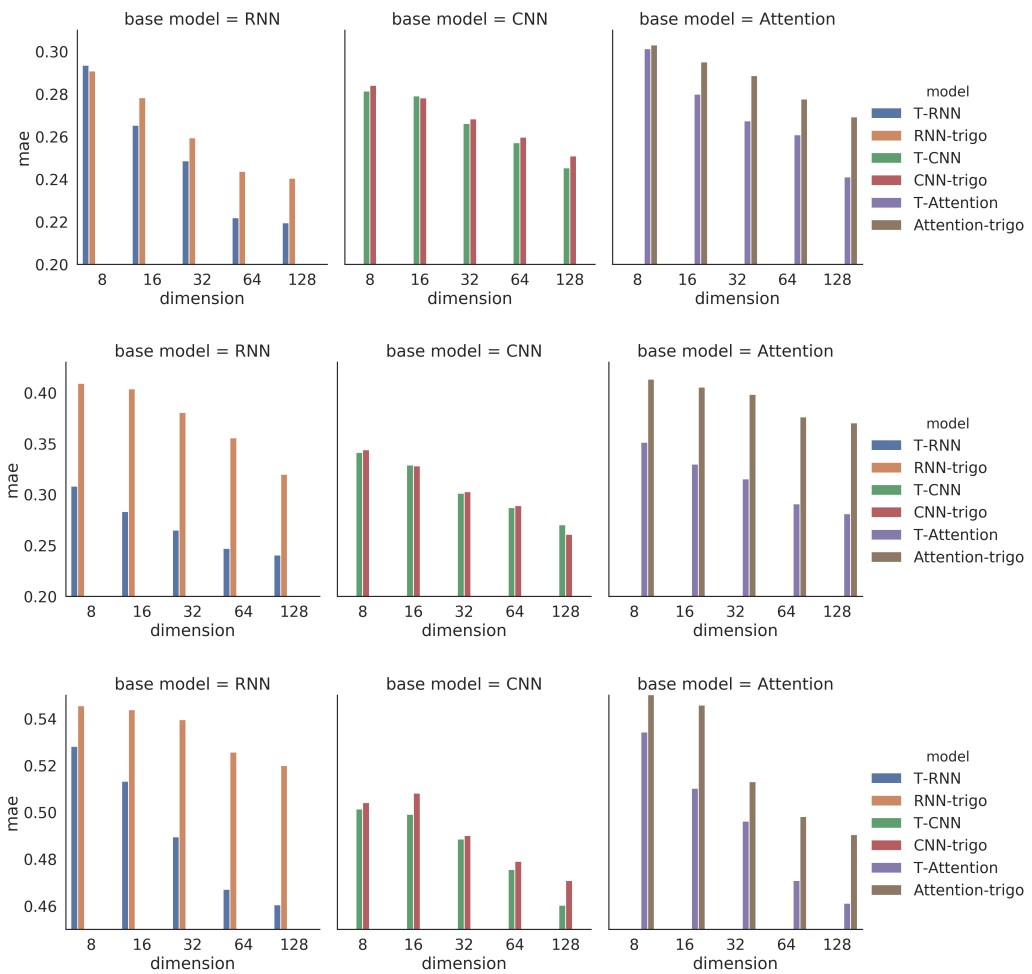

Figure A.7: Sensitivity analysis on the dimension of $\phi_\gamma$ for the Jena weather dataset. From top to bottom are the results for **Case 1**, **Case 2** and **Case 3** respectively.

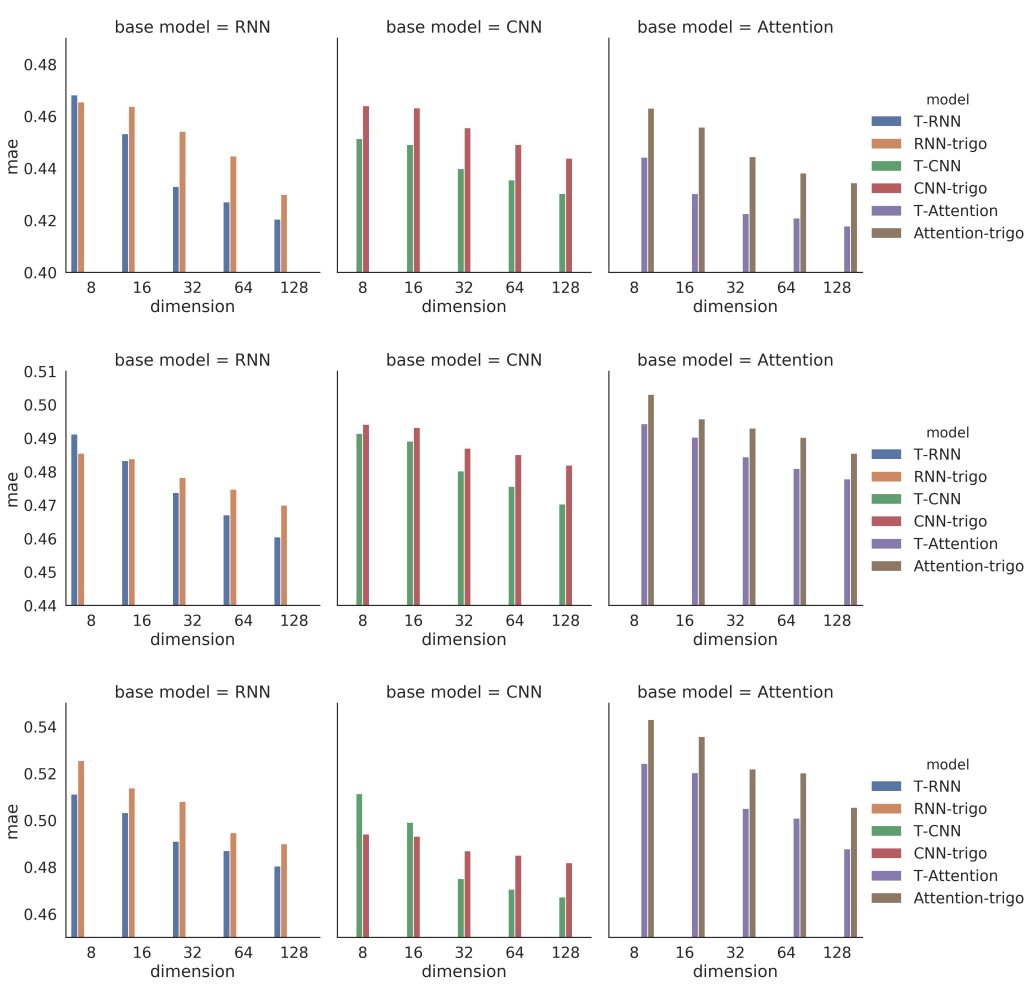

Figure A.8: Sensitivity analysis on the dimension of $\phi_\gamma$ for the Wikipedia web traffic dataset. From top to bottom are the results for **Case 1**, **Case 2** and **Case 3** respectively.

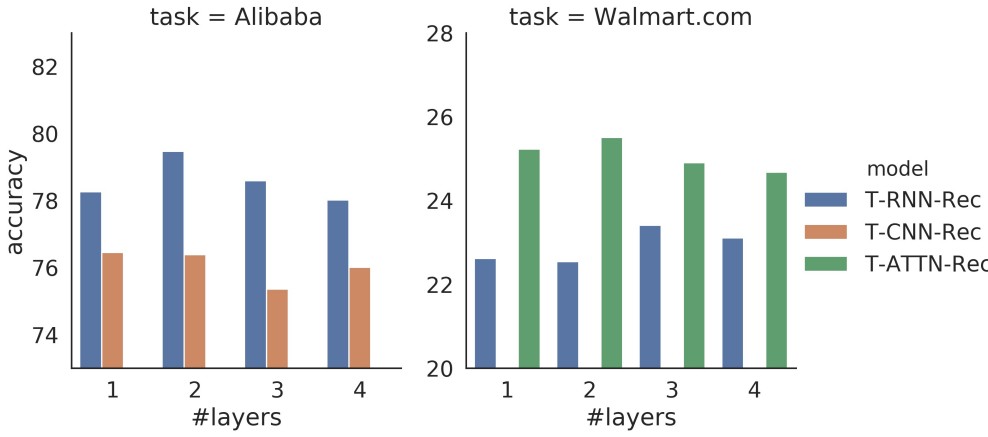

Figure A.9: Sensitivity analysis on the number of INN (affine coupling) layers for the recommendation task on Aliaba and Walmart.com datasets.

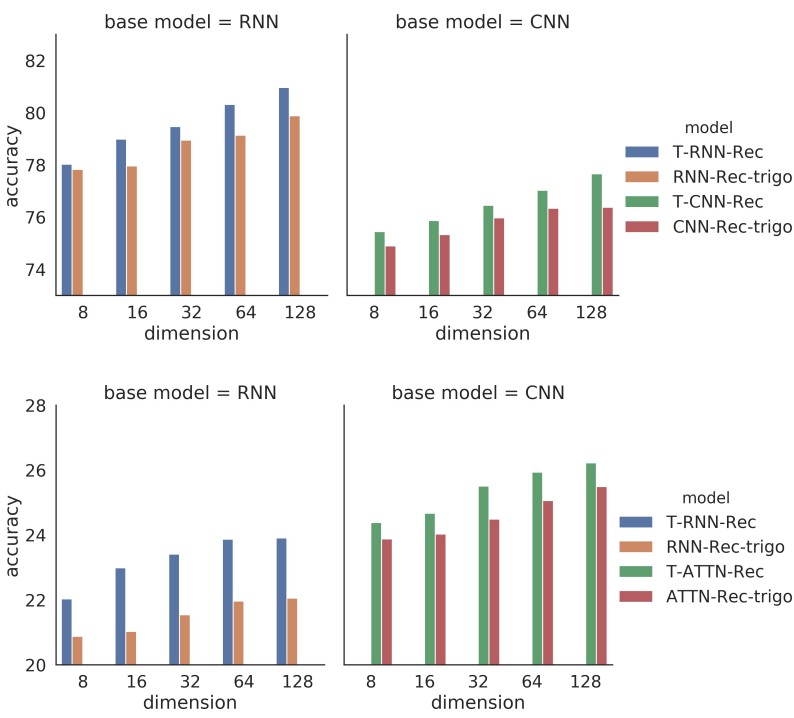

Figure A.10: Sensitivity analysis on the dimension of $\phi_\gamma$ for the recommendation task on the Alibaba dataset **upper panel** and the Walmart.com dataset **lower panel**.

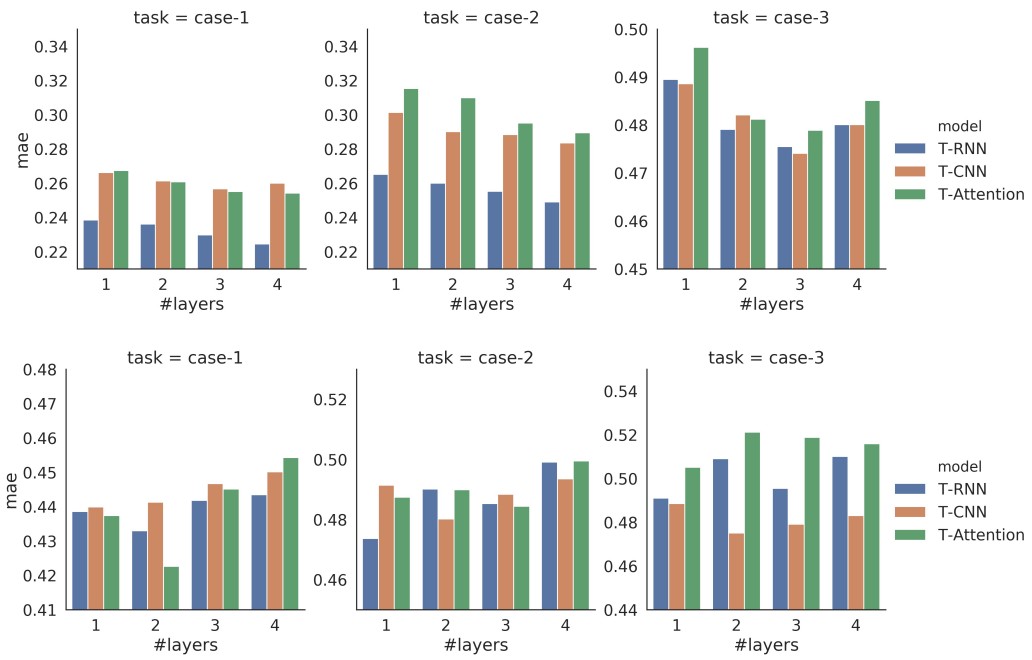

Figure A.11: Sensitivity analysis on the number of INN (affine coupling) layers. The **upper panel** gives the results on the Jena weather dataset, the **lower panel** gives the results on the Wikipedia web traffic dataset.

