# OpenReview forum: "A Temporal Kernel Approach for Deep Learning with Continuous-time Information"
_ICLR.cc/2021/Conference — ICLR 2021 Poster_

### Official Review · AnonReviewer3 · 2020-10-26
**Review from R3**

**Rating:** 7
**Confidence:** 2

**Review:**

This paper proposed a general deep learning method with temporal-kernel for continuous time series modeling.

The proposed method is technically sound and solid. The decomposition of the neural and temporal kernel brings together the kernel methods and deep learning, which delivers a general and fundamental solution to time series, especially the irregularly sampled ones or those with missing values.
In brief, this work may demonstrate a promising way of handling such problems and inspires and encourages other research in this direction.

The writing is thorough and clear. Though I did not check all proofs and the supplementary, the descriptions and arguments in the paper are properly delivered.

The proposed model consistently outperforms RNN, TCN, and attention baselines on a variety of datasets. The settings of the case 2/3 are reasonable.
Besides, it is interesting to see that the speed is still comparable to the baselines.

However, in my opinion, more baseline comparisons need to be added. I did not quite buy the claim that the proposed method is not compared with recurrent neural ODE-type models and point process models because of its more generalization and flexibility.

Minor typos:
In page 3: infitnitesimal -> infinitesimal; covaraince -> covariance
The font size in figures may be increased for better readability.

---

> ### Author Response · Authors · 2020-11-16
> **We thank the reviewer for the careful reading and providing valuable feedback**
>
> We want to thank the reviewer for the time and effort on providing valuable feedback to our paper. We apologize for the confusions caused by typos and unclear writing. We have corrected them in this version of the paper, and increased the font size in the figures. We have modified our paper according to the comments and requests by the reviewers, and all the changes we made are marked in \color{red} in this version of the paper.
>
> We agree with the reviewer that adding the neural ODE and temporal point process methods as baselines would make our arguments more solid. Given the limited time we have in the rebuttal stage, we manage to include RNN-ODE and RRN-TPP (for temporal point process) for all three cases in the time series prediction task. We choose RNN as the base architecture because the published implementations only support RNN, and extending their work to TCN and attention could be the topic of another paper. The added results are provided in Table A.1 and Figure A.2. RNN-ODE has a comparable (but slightly inferior) performance; however, the ODE solution is highly unstable, and we observe a large variance. The instability issue has also been observed and discussed by the authors of the original neural ODE paper [1].
> The RNN-TPP approach, on the other hand, does not improve the vanilla RNN baseline by much. We suspect that:
>
> 1. The model assumption of TPP is more realistic for event prediction, i.e. predicting which event (from a limited pool of events) to occur next as a classification task, rather than the standard time-series prediction;
>
> 2. The performance of TPP relies heavily on the intensity function that is very sensitive to the model specification. We have yet found a satisfactory solution from the existing literature on how to choose the intensity function.
>
> Based on the evidence, we believe that our approach also outperforms (and is more stable than) the RNN-ODE and RNN-TPP in the time-series prediction task. Unfortunately, we find it very difficult to tune the RNN-ODE and RNN-TPP model to reach a comparable performance in the recommendation task, because their model (training) architectures are somewhat restrictive. It is challenging to adapt them to a domain-specific setting. Therefore, we do not report their results for the temporal recommendation task.
>
> Also, we make the following revisions to the original paper according to the comments and requests from the other reviewers:
>
> 1. We add the dedicated algorithm description and diagram in Algorithm 1 and Figure 2. They may help the readers better understand the computation (forward and backward pass) flow of the proposed method.
>
> 2. We provide a more detailed comparison between our approach and the related work in Section 4.
>
> We again express our gratitude to the reviewer, and we are looking forward to future discussions.
>
> [1]. Duvenaud, D. K. https://www.youtube.com/watch?v=YZ-_E7A3V2w

---

### Official Review · AnonReviewer1 · 2020-10-28
**Clever treatment to decouple NN and temporal dynamics**

**Rating:** 7
**Confidence:** 2

**Review:**

The ms introduces a time component in the traditional NN setup, where the hidden layers change dynamically according to time. The idea is to borrow strength from the newly introduced time dimension to improve prediction performance.

One of the key idea is to treat each hidden layer in NN as a Gaussian process, which is represented  as “neural network kernel”. Functions drawn from this Gaussian process at different time points are assumed to follow a continuous-time system, which is actually a ODE. For some reason it is difficult to use the continuous-time system to compute the temporal kernel directly in the time domain. The ms proposes to convert the functions to the frequency domain, which leads to a nice property such that we can compute a temporal kernel (Eq. (2)) in frequency/spectral domain. Hidden layer of NN at different time points can be seen as a large Gaussian process, whose kernel could be composed by the aforementioned NN kernel and temporal kernel (Eq. (5)). This decoupling is the key of this ms.

Regarding the computation of the kernels, NN kernel can be computed by extracting the features of the hidden layer. The temporal kernel can be computed by sampling the spectral distribution, which is called the random feature representation (Eq. (7)). However, it is not clear how to specify the spectral distribution. In all examples, Normal distribution is used, which is OK but may not be able to capture the complexity of ODE.

The ms applies the proposed method to real dataset and achieve better performance than baseline methods in prediction tasks involving irregular time points setup.

In general I feel this is a nice paper. The idea of learning NN and time dynamics at the same time seems to be useful in many applications. The ms cleverly decouples the learning of NN kernel and temporal kernel in two independent modules, which can maximumly utilise current implementations.

However, due to my limited knowledge in signal processing, I am not able to dig into the mathematical details and make strong recommendations (especially Claim 2).

Some minor comments
1. What is the purpose of Claim 1? From the supplementary it just shows that f(t) and f[i] are not equal, but they may be very close and does not have a huge impact of the result. There are lots of approximations in other parts of the model.
2. Why f(t) needs to be ODE ?
3. Page 3, section 3, line 2, in the formula of f(iw), the second derivation term seems to be missing
4. Page 4, 5 lines after Eq. (3), a_2(x)!=0 => a_1(x)!=0
5. Page 5, Eq. 7, cos(tw_n) => cos(tw_m)

---

> ### Author Response · Authors · 2020-11-16
> **We thank the reviewer for the careful reading and providing valuable feedback**
>
> We thank the reviewer for the time and effort on providing valuable feedback to our paper. We apologize for the confusions caused by typos and unclear writing. We have corrected them in this version of the paper. As for the first two comments, we provide our thoughts as follow.
>
> To Comment 1: "What is the purpose of Claim 1? From the supplementary it just shows that f(t) and f[i] are not equal, but they may be very close and does not have a huge impact on the result. There are lots of approximations in other parts of the model."
>
> We think of Claim 1 more as a motivating example, rather than the main result of our paper. We agree that the discrepancy between the spectral density of f(t) and its equally-spaced discretization f[i] might be small, which ultimately depends on the regularity of the underlying continuous-time system. Here, we hope that Claim 1 can serve as a rigorous motivation for modelling the spectral density distribution: even for such a simple setting and the discretization is equally-spaced, the spectral density is changed nonetheless. As a consequence, one can only expect larger perturbations for a more complex continuous-time system under irregular sampling.
>
>
> To Comment 2: "Why f(t) needs to be ODE ?"
>
> We agree that there may be alternative options, but we find ODE the most natural characterization of a continuous-time dynamic system, and it is extensively applied for solving signal processing problems. Therefore, we hope that by exploring from the ODE perspective, our work can benefit more audience from both the deep learning and signal processing domain.
>
>
> Also, we make the following revisions to the original paper according to the comments and requests from the other reviewers, which we summarize below. All the modified parts are marked in \emph{RED} in this version of the paper.
>
> 1. We add the dedicated algorithm description and diagram in Algorithm 1 and Figure 2. They help the readers better understand the computation (forward and backward pass) flow of the proposed method.
>
> 2. We provide a more detailed comparison between our approach and the related work in Section 4.
>
> 3. In the experiments, we consider the RNN-ODE and RNN-TPP (temporal point process) as additional baseline methods for the time series prediction task. We are only able to experiment with the neural ODE and temporal point process models under RNN, because the published implementations are restricted to the RNN setting. Extending their methods to other architectures could be the topic of another paper. The results are provided in Table A.1 and Figure A.2. RNN-ODE has a comparable (but slightly inferior) performance; however, the ODE solution is highly unstable, and we observe a large variance. The RNN-TPP approach, on the other hand, does not improve the vanilla RNN baseline by much.
>
> We again express our gratitude to the reviewer, and we are looking forward to future discussions.

---

### Official Review · AnonReviewer4 · 2020-10-29
**Well written, connected with the current literature, appropriate experimental validation**

**Rating:** 7
**Confidence:** 3

**Review:**

This article proposes a methodology to *adapt* NNs to continuous-time data though the use of a (temporal) reproducing kernel. I enjoyed reading the paper, the message is clear, illustrative and the connection with other existing works is to the point. Although I am unfortunately unable to assess the theoretical novelty of the paper (I am unaware of the details of the state of the art in the subject) the contribution of the paper relates to the study of a kernel, given by an ODE, attached to the input of the NN. This kernel is also represented using Fourier feature expansions.

Though the paper heavily relies on well-known concepts  (standard NN, GPs, Fourier features), I see that is has a contribution.

I suggest the following amendments:
-for some readers, the general proposed architecture might be confusing. Perhaps a diagrams (similar to that in Fig A.1) would be useful in the first pages of the paper. How does the kernel turn the continuous-time data into NN-ready?
-much useful material is relegated to the appendix, if key results, scope and more are only in the appendix, they might not receive the deserved attention.
-please better clarify how different your work is from the existing literature: NTK, deep kernel learning, neural ODEs, etc

---

> ### Author Response · Authors · 2020-11-16
> **We thank the reviewer for the careful reading and providing valuable feedback**
>
> We want to thank the reviewer for the time and effort on providing valuable feedback to our paper, particularly for suggesting the amendments. We believe they help bring a better version of our paper.
> We have modified our paper according to the comments and requests by the reviewers, and all the changes we made are marked in \color{red} in this version of the paper.
>
> In response to the comments:
>
> To Comment 1. "For some readers, the general proposed architecture might be confusing."
>
> We thank the reviewer for pointing out the potential confusion in terms of the architecture and computation workflow. We add a detailed diagram in Figure 2, together with the algorithm description for the forward and backward pass in Algorithm 1. We believe they help clarify the workflow of our approach.
>
> To Comment 2. "How does the kernel turn the continuous-time data into NN-ready?"
>
> The key for converting the continuous-time data to NN ready is to use the random feature representation, which can be thought of as a time encoding functional that takes the continuous-time data as input, and produce a vector representation. We provide a detailed illustrate on this conversion in Figure 2.
>
> To Comment 3. "Some useful material is relegated to the appendix."
>
> We agree that some of the material should be moved to the main paper. In particular, we add the scope and limitation to Section 6, and the related-work comparisons to Section 5. As for the numerical results, since other reviewers have requested for additional experiments, we decide not to change the original layout and presentation in the rebuttal stage to avoid confusions. We will move more results to the main paper in the final version.
>
> To Comment 4. "Please better clarify how different your work is from the existing literature: NTK, deep kernel learning, neural ODEs, etc."
>
> Our approach characterizes the continuous-time ODE system via the lens of kernel. It complements the existing neural ODE methods which are often restricted to specific architectures, rely on ODE solvers and lack theoretical understandings.
> The existing work on deep kernel learning mostly assumes a given kernel family. We propose a novel deep kernel learning approach by parameterizing the spectral distribution under random feature representation, and demonstrate its effectiveness with both theoretical arguments and numerical results.
> Finally, since the idea of composing NN with a temporal kernel under its random feature representation is proposed in this paper for the first time, the derivations of the associated NTK is novel in its own regard.
>
> We again express our gratitude to the reviewer, and we are looking forward to future discussions.

---

> > ### Comment · AnonReviewer4 · 2020-11-23
> > **I have read the rebuttal**
> >
> > Dear authors, I have read the rebuttal. Many thanks for the explanation.
> > I have increased my review based on your response.

---

### Official Review · AnonReviewer2 · 2020-10-30

**Rating:** 6
**Confidence:** 3

**Review:**

##### Post-rebuttal update

I've read the rebuttal and updated my score.

---------------

This paper proposes a deep learning model for incorporating temporal information by composing the NN-GP kernel and a temporal stationary kernel through a product. The temporal stationary kernel is represented using its spectral density, which is parameterized by an invertible neural network model. This kernel will be learned from the training data to characterize its temporal dynamics.

##### Originality & Significance
The modeling approach taken in this paper is original to my best knowledge. Although it is well-known that second-order stationary processes has a SDE correspondence, it is rare that this property is connected to NN as GPs and this work finds an application where such ideas can be potentially useful. However, I find it difficult to say anything about significance of this idea since it is not very clearly described. I encourage the authors to make a substantial revision to clarify the issues listed below.

##### Clarity
The clarity is low. Although the motivation and the high-level idea is clear, I find it very difficult to understand the actual approach taken by this work. There is no description of the actual algorithm and I can see many algorithmic and computational issues left without discussion:
* How is prediction made at a specific (t, x)? Do you use a GP predictive mean conditioned on the training points?
* If the prediction is made by GPs, how do you solve the scalability issue? When the training set is large, do you take a sparse approach? The temporal kernel is defined through a random feature representation, do you take advantage of it for fast computation?
* or you just take a weight-space approach and compose the features (take pairwise product of the features of k_T and \Sigma to form the new features)?
* Is NN-GP or NTK kernels used to compute the kernels? How do you compute them? Do you use a Monte-Carlo estimate or the closed-form (computed through a recursion)?
I will be happy to raise the score if these questions are properly addressed.

##### Strengths
* The modeling approach is novel.
* The proposed method consistently outperforms other baselines in handling irregular continuous-time data.

##### Weaknesses
* The method used is not clearly described.
* The non-stationary extension to Bochner's theorem is a known result.
* Although the performance is shown to outperform other NN-based approaches in experiments, there might be scalability issues to apply the approach to larger-scale problems with long sequences (assume a non-weight-space approach).

##### Minor
P16: A.4.1: "We the Fourier transformation transformation on both sides"?

---

> ### Author Response · Authors · 2020-11-16
> **We thank the reviewer for the careful reading and providing valuable feedback**
>
> We want to thank the reviewer for the time and effort on providing valuable feedback to our manuscript. We believe the comments help us to bring a better version of this paper. We have made substantial modifications to the original paper, and all the changes are marked in RED in this version of the paper.
>
> We first apologize for the typos and unclear writing. We have corrected them in the paper.
> As for the clarity issues of the algorithm and computation, we add a detailed diagram to illustrate the architecture in Figure 2, and provide the algorithmic description for the forward and backward computation in Algorithm 1. They help clarify the questions from the reviewer, which we also summarize as below.
>
> Question 1. "How is the prediction made at a specific (t, x)? Do you use a GP predictive mean conditioned on the training points?"
>
> The detailed answer is provided in Algorithm 1. In short, we first sample from the auxiliary distribution and use the invertible neural network to parameterize the random samples. We then construct the random features as in eq. (8) as a time-aware encoding (which is exactly the random feature representation of the temporal kernel), and combine the encoding with the selected hidden layer of the neural network. Our strategy for the forward passing (prediction) is similar to that of the variational autoencoder[1], which saves us from the computation complexity of using GP to make predictions.
>
> Question 2. "If the prediction is made by GPs, how do you solve the scalability issue? When the training set is large, do you take a sparse approach? The temporal kernel is defined through a random feature representation, do you take advantage of it for fast computation?"
>
> Following our answer to the previous question, and we hope the reviewer could take a glance at the new Figure 2 and Algorithm 1, since we are not using GP to make predictions, the computation complexity is almost the same as the original neural network. The computing time comparisons are actually provided in Figure A.5 (we understand it is not mandatory for the reviewers to check the appendix), so we mention it here that the training and inference computation time are almost the same for CNN, RNN and attention mechanism using our approach. The trick for speeding up the computation is precisely by taking advantage of the random feature representation, as pointed out by the reviewer, where we use the reparameterization to construct the representation under the INN efficiently.
>
> Question 3. "Do you just take a weight-space approach and compose the features"
>
> All the operations are conducted in the feature (weight) space, i.e. when doing training and inference, instead of computing the kernel compositions, we conduct the corresponding compositions in their feature spaces.
>
> Question 4: "Is NN-GP or NTK kernels used to compute the kernels? How do you compute them? Do you use a Monte-Carlo estimate or the closed-form?"
>
> Both the NN-GP and NTK kernels can be computed, though we do not explicitly rely on their outcome in the training and inference, by Monte-Carlo estimate via sampling from the auxiliary distribution. The output from the RF-INN module in Figure 2 can now be used to construct the random feature representation, from which we can recover the NN-GP kernel (via Claim 1) and NTK kernel (via Proposition A.1). We agree that it could be an exciting topic to study the properties of the obtained NN-GP and NTK kernel, which we leave to future work.
>
> We also wish to justify the weakness commented by the reviewer.
>
> Comment 1: "The method used is not clearly described."
>
> We provide the detailed descriptions of our approach in the new Figure 2 and Algorithm 1.
>
> Comment 2: "The non-stationary extension to Bochner's theorem is a known result."
>
> We have discussed in the appendix that we refer to the Yaglom Theorem [2] and the seminal work of [3] to construct the random feature representation of non-stationary kernels. However, our stochastic convergence result in Proposition 1 is novel to the best of our knowledge. We agree with the reviewer that our claim about the novelty in terms of the non-stationary kernel is not precise, and we will modify it to focus on the theoretical contributions.
>
> Comment 3: "There might be scalability issues to apply the approach to larger-scale problems with long sequences"
>
> As we have clarified in the previous response, the scalability issue is not a concern for our approach. As we show in the new Figure 2, the computation complexity with respect to sequence length is at the same order compared with the vanilla RNN (for example), since we conduct the sampling and reparameterization with the invertible neural network at the beginning of each batch.
>
> [1]. Kingma D P, Welling M. Auto-encoding variational bayes, 2013
>
> [2]. Yaglom A M. Correlation Theory of Stationary and Related Random Functions, 1987
>
> [3]. Remes S, Heinonen M, Kaski S. Non-stationary spectral kernels, 2017

---

### Decision · Program_Chairs · 2021-01-07
**Final Decision**

**Decision:**

Accept (Poster)

**Comment:**

This paper presents a novel approach for integrating time into deep neural network models based on the Gaussian process limit view of a neural network model. Specifically, the approach augments an a-temporal neural network designed to process a single time point with a temporal kernel that relates data points across time. The composition of the a-temporal neural network kernel with with the temporal kernel is accomplished efficiently using a random features representation of the temporal kernel. The authors propose to represent the temporal kernel via its spectral decomposition, which makes the approach quite flexible. Learning leverages re-parameterization. While random features have been used to approximate temporal kernels in prior work [1], the approach in this paper is significantly more general in that it can be composed with any a-temporal deep architecture and the authors show results for RNNs, CNNs, and attention-based models. The predictive performance of the approach also appears to be consistently better than baselines and it works particularly well on the challenging case of irregularly sampled data.

In terms of weaknesses, the reviewers had a number of questions about the paper. The authors updated the paper to include some more recent models including ODE-RNNs. This material is currently presented in the appendices and needs to be moved into the main paper. Several of the reviewers also had technical questions questions that are in fact addressed in the manuscript; however, the authors are relying heavily on the appendices to present many important details and the paper is currently over 30 pages long. The frequent references to the appendix for additional details makes the paper a challenging read. The authors have already done some work to address clarity by adding a new figure, but should prioritize moving additional key details into the main body of the paper to improve readability.

[1] http://auai.org/uai2015/proceedings/papers/41.pdf